# PTR: A Benchmark for Part-based Conceptual, Relational, and Physical Reasoning

**Yining Hong** [*]
UCLA

**Li Yi**
Stanford University

**Joshua B. Tenenbaum**
MIT BCS, CBMM, CSAIL

**Antonio Torralba**
MIT CSAIL

**Chuang Gan**
MIT-IBM Watson AI Lab

## Abstract

A critical aspect of human visual perception is the ability to parse visual scenes into individual objects and further into object parts, forming part-whole hierarchies. Such composite structures could induce a rich set of semantic concepts and relations, thus playing an important role in the interpretation and organization of visual signals as well as for the generalization of visual perception and reasoning. However, existing visual reasoning benchmarks mostly focus on objects rather than parts. Visual reasoning based on the full part-whole hierarchy is much more challenging than object-centric reasoning due to finer-grained concepts, richer geometry relations, and more complex physics. Therefore, to better serve for part-based conceptual, relational and physical reasoning, we introduce a new large-scale diagnostic visual reasoning dataset named PTR. PTR contains around 70k RGBD synthetic images with ground truth object and part level annotations regarding semantic instance segmentation, color attributes, spatial and geometric relationships, and certain physical properties such as stability. These images are paired with 700k machine-generated questions covering various types of reasoning types, making them a good testbed for visual reasoning models. We examine several state-of-the-art visual reasoning models on this dataset and observe that they still make many surprising mistakes in situations where humans can easily infer the correct answer. We believe this dataset will open up new opportunities for part-based reasoning. PTR dataset and baseline models are publicly available [2].

## 1 Introduction

A long-standing challenge in the field of artificial intelligence is to enable machines to reason and answer questions about visual scenes. Several datasets [64, 16, 29, 49, 57] have been proposed to tackle this challenge. They mostly focus on object-level features without emphasizing much on detailed part-level understanding. However, there is strong psychological evidence that human beings parse visual scenes into part-whole hierarchies (*e.g.,* from a scene to the objects, from an object to its parts), which are currently missing from existing datasets.

Inspired by Aristotle's quote "the whole is greater than the sum of its parts", there has been a long history of research in psychology concerning how parts and wholes are related, beginning with the Gestalt psychologists [47]. There have also been recent efforts on how to represent such part-whole hierarchies using neural networks [19]. Introducing part-whole hierarchies into visual reasoning

---

[*]The work was done while Yining Hong was a research intern at MIT-IBM Watson AI Lab.
[2]Project page: http://ptr.csail.mit.edu/

35th Conference on Neural Information Processing Systems (NeurIPS 2021), virtual.

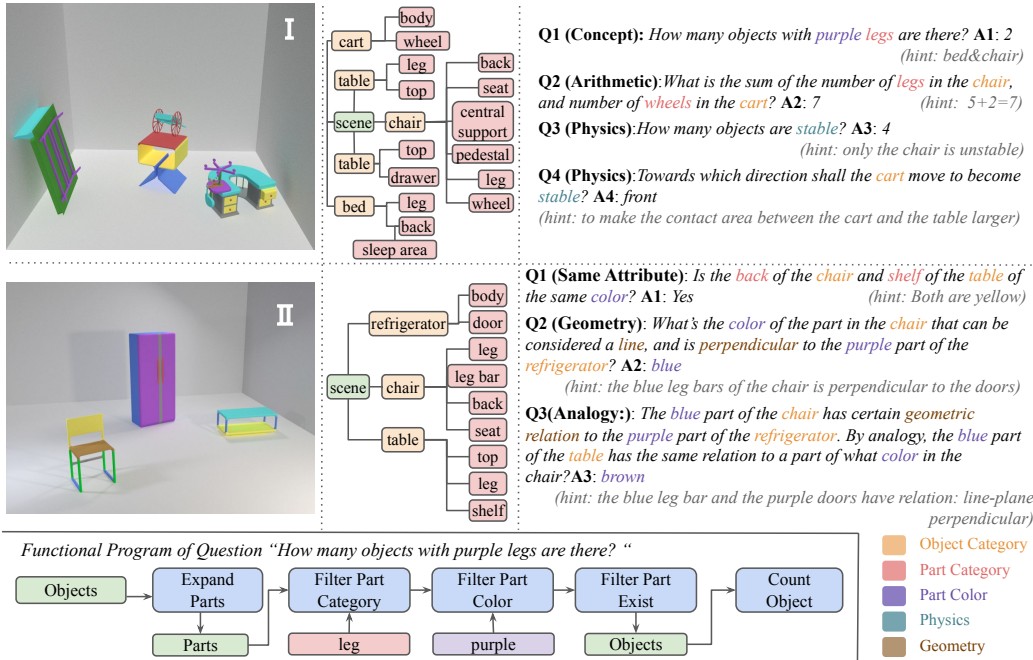

Figure 1: Exemplar scene graphs, questions, and programs of our Part Reasoning (PTR) dataset. PTR contains five question types: concept, relation, analogy, arithmetic and physics. Top left shows scenes from our dataset, paired with hierarchical scene graphs in the middle. Top right presents questions, answers and hints to the answers. A functional program of a question is shown in the bottom.

brings two unique challenges. The first is how to discriminate objects using part-level attributes. Unlike previous works which refer to objects based on holistic attributes such as object category, the specific ontology nature of objects and their parts requires a detailed study. The second is how to leverage part-level attributes to interconnect objects. Objects of various categories are interrelated via some common parts (*e.g.*, leg, central support, pedestal). The reasoning performed on one category should thus be easily generalized to reasoning on unseen categories with shared parts. Human beings are endowed with such modular but interconnected perceptual systems. They can effortlessly discriminate between the two tables in Figure 1 [I] based on the differences in the tops and drawers, as well as observe a connection between the bed and chair due to the similarity of legs. It remains to be explored whether machines have the same hierarchical perceptual capabilities.

Based on the highly composite part-whole hierarchies, visual reasoning tasks can be richer, more complex, as well as more challenging. First, the introduction of parts enriches the diversity of both visual perception and question understanding. Apart from basic object-level properties such as color and category, the visual scenes and natural language questions extend upon the properties of object parts, the composition of which makes the image-question pairs more distinctive. Second, the relations between parts go beyond simple spatial relations. Parts can often be approximated as oriented geometric primitives (*e.g.,* line, plane), and there naturally exist abundant geometric relations between these primitives such as "perpendicular" and "parallel" (*e.g.,* Q2 in Figure 1[II]). Analogical reasoning can also be established given such relationships. Third, reasoning on parts enables the understanding of implicit properties of objects, such as physics. For example, the arrangements of the parts and their geometric relations affect the stability of the objects (*e.g.,* Q3, Q4 in Figure 1[I]).

In this paper, we present a large-scale ParT Reasoning dataset, or PTR for short, a benchmark for part-based conceptual, relational and physical reasoning. It includes ∼ 70k scenes paired with 700k questions containing part-whole structures. We include over 10k objects from the PartNet [45] dataset across five object categories (chair, table, bed, refrigerator, cart) with rich geometric and structural variations to construct our scenes. Our dataset consists of five types of questions: concept, geometry, analogy, arithmetic, and physics. We provide scene graph annotations including the ground-truth locations, segmentations, and properties for all objects and parts, as well as all the object-object

| Dataset | 3D | Objects | Parts | Relation | Diagnostic | Geometry | Analogy | Physics |
|---|---|---|---|---|---|---|---|---|
| VQA [3] | ✓ | ✓ | ✗ | ✓ | ✗ | ✗ | ✗ | ✗ |
| VCR [61] | ✓ | ✓ | ✗ | ✓ | ✗ | ✗ | ✗ | ✗ |
| GQA [28] | ✓ | ✓ | ✗ | ✓ | ✓ | ✗ | ✗ | ✗ |
| CLEVR [29] | ✓ | ✓ | ✗ | ✓ | ✓ | ✗ | ✗ | ✗ |
| RAVEN [62] | ✗ | ✓ | ✓ | ✓ | ✓ | ✗ | ✓ | ✗ |
| PTR (ours) | ✓ | ✓ | ✓ | ✓ | ✓ | ✓ | ✓ | ✓ |

Table 1: Comparison between PTR and other visual reasoning benchmarks.

and part-part relationships for model diagnostic. We also provide functional programs paired with questions.

We analyze a suite of state-of-the-art visual reasoning models on the PTR dataset and find that they all struggle with it, especially in relational, analogical, and physical reasoning. One oracle neural symbolic model performs better than other models but highly relies on additional supervision such as object and part masks and visual attributes from simulations. Also, the performances of all the models are inferior to human performance by a large margin, suggesting that there's still a long way to go to equip machines with human-like hierarchical perceptual and reasoning abilities.

## 2 Related Work

### 2.1 Visual Reasoning

To assess machines' ability on understanding and reasoning over vision and language, a wide variety of benchmarks have been created over the recent years [29, 3, 28, 16, 61, 12, 64, 36]. In terms of data input, our dataset mostly resembles the CLEVR dataset [29], which is also a diagnostic dataset of synthetic images, template-based questions and compositional programs. However, the objects in CLEVR's scenes are of simple shapes and do not contain parts to form complex logical chains of reasoning. The VQA dataset [3] contains large-scale crowd-sourced real images and human-generated questions. Since it's not a fully-controlled synthetic dataset, it shows great inherent bias. Visual Commonsense Reasoning (VCR) [61] focuses on reasoning on commonsense knowledge. The GQA dataset [28] pairs real images with synthetic compositional questions. However, none of these datasets provide part-based dense annotations and questions. RAVEN [62] associates vision with structural, relational, and analogical reasoning in a hierarchical representation. The dataset asks models to choose an image according to analogy. However, their images are rather simple, mainly consisting of 2D shapes like circles and triangles, and they do not pair images with natural language questions.

Although numerous visual reasoning datasets have been proposed, the reasoning chain typically terminates at the object level for these datasets, due to the mere reference to objects based on holistic features or spatial relationships. Also, while previous datasets emphasize reasoning based on spatial relationships, humans possess diverse reasoning abilities, including analogy [62], geometry[48, 40], arithmetic[63, 20, 21], and physics [13]. These aspects can be effortlessly incorporated into visual reasoning to constitute a benchmark that further approaches humans' intellectual level. Thus, we create a new visual-reasoning dataset that focuses on multiple aspects of reasoning on part-whole relations. We compare our dataset with previous visual reasoning datasets in Table 1.

A large number of visual reasoning models have been proposed. Specifically, MAC [27] combined multi-modal attention for compositional reasoning. LCGN [23] built contextualized representations for objects to support relational reasoning. These methods, though embedded in neural networks, model the reasoning process implicitly. Neural-symbolic methods [56, 42, 7, 38] explicitly perform symbolic reasoning on the objects representations and language representations. Recently, methods based on transformers [33] have achieved remarkable performance on visual reasoning tasks by utilizing end-to-end object detector. In this paper, we conduct experiments and assess strengths and weaknesses of these baseline models on part-based visual reasoning.

## 2.2 Part-based 3D Understanding

Understanding 3D parts has been a long-standing problem in computer vision and graphics. It is important due to several reasons. Firstly, parts and their arrangements that describe the detailed object geometry and structure are helpful for object discrimination purposes [9, 1]. Secondly, parts are usually action handles and their forms could reveal the object functionality [34, 43, 24], therefore crucial for interacting with objects. Thirdly, functionally related object parts usually share similar forms, making parts suitable for connecting different objects, facilitating knowledge transfer and promoting generalizable learning [17, 41]. Finally, parts are often commonly used for 3D generation and content editing purposes [11, 44, 46].

To facilitate part-based 3D understanding, various part-centric 3D datasets have been proposed in the literature. ShapeNet part dataset [58] annotates 31,963 shapes from ShapeNet [4] with semantic part segmentation covering 16 categories. And each shape is annotated with 2 to 5 parts. PartNet [45] further increases the granularity of parts, contributing 573,585 finegrained part instance annotations for 26,671 shapes across 24 object categories. Going beyond semantic understanding, several datasets also focus on the functionality and articulation aspects of parts [54, 43, 24, 25, 52]. However, most of these datasets target at part identification rather than part-based conceptual, relational and physical reasoning. Chang et al. [5] collected 27,477 part instances from 2,278 models covering 90 object categories, with a focus of studying conceptual part-based reasoning. ShapeGlot [1] presents an object reference dataset with part-level conceptual reasoning implicitly considered. Want et al. [51] proposes PartNet-Chairs, which contain questions about parts of chairs. These datasets are restricted to conceptual reasoning while our PTR has a broader scope covering ampler reasoning types.

Restricted by the annotations in existing large-scale datasets, previous part-based 3D understanding works mainly focus on identifying parts [15, 30, 26, 55, 32, 58] and part relations [6, 10, 31, 35, 60, 50] for individual 3D objects. Several methods also consider a more complex hierarchical structure of 3D parts [59, 44, 37, 53, 14, 22, 8]. Moreover, these works usually assume a strong categorical prior and tend to have poor generalizability across categories. However, we human beings can easily do more than just part identification or single-object part relationship understanding for objects from known categories. We can reason the relationship between parts from different object instances, make analogies, conduct mental arithmetic, and even infer the physical realism or stability. These tasks are not well supported by current datasets. Therefore we present PTR to close this gap and to better support the study of these human-like common sense.

## 3 The PTR Dataset

The PTR dataset aims to assess machines' ability to perform part-based visual reasoning. We carefully design this dataset in a fully-controlled synthetic environment and enable model diagnostics on this complex reasoning task. Each image has ground-truth annotations regarding the locations, orientations and attributes of objects and parts, as well as the object masks and part masks. We also provide depth images, camera locations and orientations to facilitate the possible use of 3D models on our dataset in the future. For reasoning on geometric relations of parts, we include the geometric information of a part indicating whether it can be considered as a geometric line or plane, and the line equation or plane equation. For physical reasoning, we include the stability information of each object. The images are paired with five types of questions: concept, relation, analogy, arithmetic and physics. The questions have associated functional programs.

### 3.1 Image Generation

PTR includes approximately 52k images for training, 9k for validation and 10k for testing. The images are rendered via Blender[3]. To get the physical information, we apply Bullet[4] to simulate the future motion traces of each object.

**Scene, Objects and Parts.** The PTR universe contains five categories of objects (chair, table, bed, refrigerator and cart) from the PartNet dataset. The reasons why we choose these categories are as follows: 1) These are commonly seen objects in real life scenes; 2)The objects share a lot of generic

---

[3]https://www.blender.org/
[4]https://pybullet.org/wordpress/

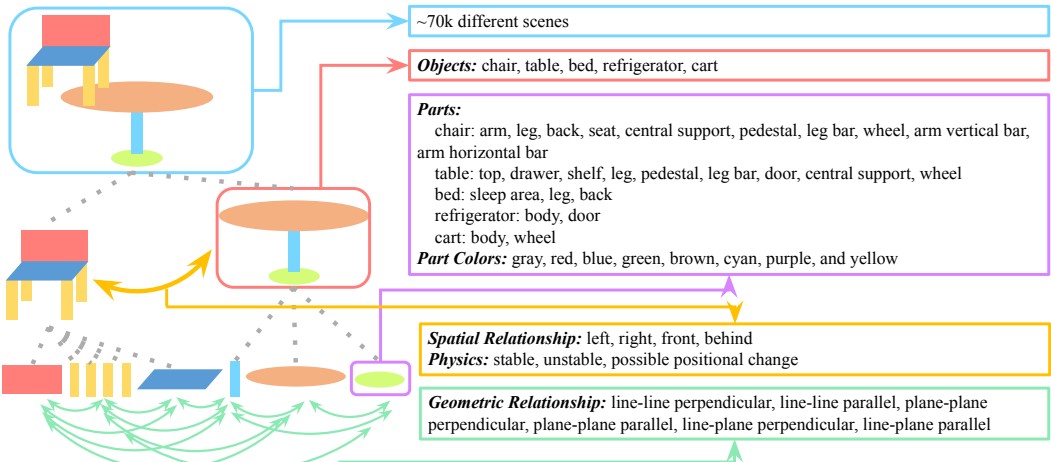

Figure 2: An overview of the scene structure and concepts in the PTR dataset. The PTR universe contains five object categories, each consisting of a set of part categories. Objects have spatial relationships and parts have geometric relationships. Physical properties are also defined on objects.

parts (*e.g.,* leg, leg bar, central support, pedestal, door, wheel, *etc*.); 3)The objects have rich physical configurations depending on the constitution of the parts. Each object can have an arbitrary number of parts. Each part takes one of eight colors (gray, red, blue, green, brown, cyan, purple, and yellow). The specific concepts used in the PTR dataset are listed in Figure 2. The distribution of the number of parts in the scenes is shown in Figure 3c.

When constructing the scene, we first place a floor and three walls of random texture to enrich geometric and physical information. We add jitters to the locations and orientations of the camera and the lamps. Every scene has three to six objects. To place an object, we ensure that no objects overlap (except for physical scenes which have objects stacking). We also ensure that the objects are of proper orientations so that no much burden would be placed on object detection. The training dataset and test dataset has no shared 3D object models (*i.e,*, the PartNet 3D models are not shared across splits, but the semantics are shared).

The physical scenes contain objects that are tilted, leaned towards walls, or stacked onto other objects. We use the Bullet physics engine to simulate physical effects. We calculate the change of object locations and orientations along time steps to determine the stability of an object. If the change of location and orientation is within a threshold value, an object can be considered stable. For unstable objects, we move the objects in four directions (front, behind, left, right) around their original location, to account for possible changes for the objects to become stable. To avoid missing such possible changes, for each direction we define four moving distances to place the objects.

**Relationships** PTR has three types of relationships: spatial relationships of objects, geometric relationships of parts, and same-attribute relationships of both objects and parts.

Objects are spatially related via four relationships (left, right, in front of, behind). The spatial relationships are dependent upon the camera viewpoint.

Parts have geometric relations if they, with negligible thickness, can be considered as 1D lines or 2D planes (*e.g.,* the legs and leg bars of Figure 1 [II] can be treated as lines, and the doors can be treated as planes). To detect such geometric primitives, for each part we randomly sample two points for lines, or three points for planes from the raw 3D data, for 1000 times. Each time we derive a line equation or plane equation from the sampled points. We ensure that distances between the points are sufficient enough with regard to the size of the part to reduce noise. The final line or plane equation is then approximated by averaging the equations. We then further sample 2500 points from the 3D data to see if the majority of points conform to the equation. If not, the part cannot be considered as a line or plane. The same-attribute relationships involve same-category relationship between objects, as well as same-category and same-color relationships between parts.

## 3.2 Question Generation

We pair each image with machine-generated questions coupled with executable functional programs. PTR contains approximately 520k questions for training, 90k for validation and 100k for testing. All questions are open-ended and can be answered with a single word. Sample questions and detailed distributions can be found in Figure 1 and Figure 3.

**Concept** Conceptual questions evaluate a model's capability to understand and reason about basic part-whole relations. The reasoning tasks are grounded within the compositional space of both object-level and part-level properties. Unlike previous visual reasoning datasets which refer to an object based on the object's holistic attributes, PTR distinguishes objects by the existence of parts and the attributes of parts. To shed light on the interconnected nature of part-whole hierarchies, there are also questions that do not specify any object category but only the common part categories (*e.g.,* Q1 in Figure 1 [I] makes a connection between bed and chair by asking about the objects with legs). Conceptual questions contain multiple sub-types including: *query object category*, *query part category*, *query part color*, *count object*, *exist object* and *count part*.

**Relation** Relational questions query about the spatial relationships of objects, geometric relationships of parts, as well as same attribute relationships of objects and parts. Upon querying about the spatial relationships, the questions take the six sub-types of the conceptual questions, except that objects are also filtered using spatial information.

As for geometric relationships, we first detect a part that can be considered as a line or a plane, refer to the part using color, and then query the existence/count/color of parts in another object that has certain geometric relation to this part. There are in total six types of geometric relationships, which are listed in Figure 2. Q2 in Figure 1 [II] is an example of geometric questions. By considering the blue leg bars as lines and the purple doors as planes, we can easily examine that the two parts are perpendicular.

The same attribute relationships take three forms. The first one asks the model to compare two attributes of two objects/parts and return a "yes/no" answer (*e.g.,*, Q1 of Figure 1 [II]). The second refers to an object with the "same-category" relation to another object. The third detects whether an object has the same part attributes (*e.g.,* same color of legs) as another object.

**Analogy** Analogical questions query analogy on spatial relationships and geometric relationships. Specifically, given objects or parts A, B, C where B has a certain relation to A, the model is required to select an object or part D that has the same relation to C. The question may query about the existence/count/attribute of D. An exemplar question is shown in Figure 1 [II] Q3. In this question, A is the leg bars of the chair, B is the door of the refrigerator. We find that the two parts have the "line-plane perpendicular" relationship. C is the legs of the table, and the question queries about the color of D, such that C and D have the same relation as A and B.

**Arithmetic** Arithmetic questions take the quantities of two types of parts in two objects, and apply two types of operations to the quantities. The first type is "compare integer". The model is asked to predict whether the two quantities follow "equal/greater than/less than" relationships and return a "yes/no" answer. The other type is "sum/minus"", where the sum or the difference of two numbers is queried (*e.g.,* Figure 1 [I] Q2). We ensure that the answer is no greater than 10 or less than 0 to facilitate the training of neural models.

**Physics** Physical questions query about the stability of objects. For example, Q3 in Figure 1 [I] asks how many objects are stable. If an object is unstable, further questions can be asked concerning the possible changes to the location of the object to make it stable, as in Figure 1 [I] Q4.

**Program** In PTR, each question can be parsed into a hierarchical tree-structured functional program, as shown in the functional programs in Figure 1. Different from previous datasets, PTR has three key functions: *expand parts*, *filter part exist* and *filter part count*, which enable the flexible transition between object-level reasoning and part-level reasoning. When the reasoning procedure switches from object level to part level, *expand parts* takes all the objects, and returns the parts of the objects. *filter part exist* filters objects by examining whether certain parts exist in the objects, and *filter part count* filters objects by the number of certain parts in objects. A list of program modules can be found in the Supplementary Material.

**Bias control** Questions are generated from pre-defined templates, and transformed into natural language questions with associated object-level and part-level attributes from the scene. We manually

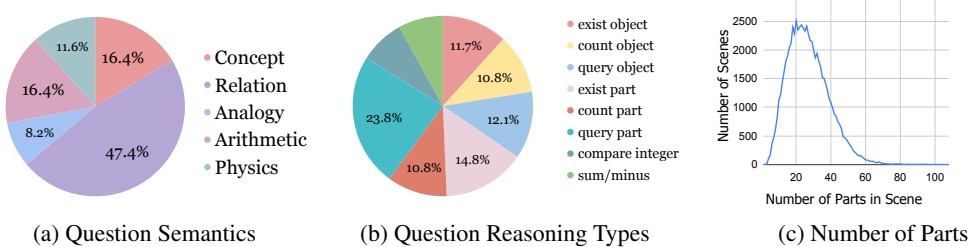

| (a) Question Semantics | (b) Question Reasoning Types | (c) Number of Parts |

Figure 3: Analysis on question types and distributions.

define 58 templates for question generation. We sort the question types and templates each time we generate a question to ensure a balanced question distribution. To avoid bias, we force a flat answer distribution for each question type by rejection sampling. Specifically, when sampling a question-answer pair, if the count of the answer exceeds the median count of all answers in that type by a margin, we discard the question-answer pair.

## 4   Baseline Evaluations

### 4.1   Baseline Models

We mainly evaluate three kinds of state-of-the-art visual reasoning methods, including heuristic models, end-to-end neural networks, and neural-symbolic models.

**Q-type (Rand.), Q-type (Freq.)**: These are language-only baselines that examine the bias in the dataset. Q-type (Rand.) uniformly samples an answer from the answer set given a specific question type. Q-type (Freq.) selects the most frequent training set answer for each question type.

**UP-DOWN**: We extract part-centric features in an unsupervised manner using Slot Attention[39], and apply bottom-up and top-down attention [2] on the part-centric features.

**CNN+LSTM** The question is encoded by the final hidden states from a Long Short Term Memory network (LSTM) and the image is encoded using features from a convolutional network (CNN). The features are concatenated and fed to an MLP to predict the final answer. This is a simple baseline that examines how vanilla neural networks perform on PTR.

**MAC, MAC(P)** ([27]) MAC utilizes a novel Memory, Attention and Composition cell to perform iterative reasoning process. The image features and question features are incorporated via a joint attention mechanism. This is a strong baseline achieving state-of-the-art performance on CLEVR without any supervision on vision and language. To boost part-level reasoning, we construct part-aware features by adding the segmentation masks of all parts and their attributes to the original features, denoted as MAC(P).

**LCGN** ([23]) The objects are represented as nodes in the graph network, and are described by a context-aware representation from related objects through iterative message passing conditioned on the textual input.

**MDETR** ([33]) This model fuses the two modalities at an early stage, leveraging an end-to-end modular detector to boost downstream tasks like visual question answering. It uses ground-truth bounding boxes of parts to train the detector.

**NS-VQA** ([56]) The visual recognition and language understanding are performed by neural networks, which are passed to a symbolic execution module to get the final results. We treat this baseline as an oracle model where we use the annotations of both vision (ground-truth masks) and language (ground-truth programs).

**Implementation Details** We use an ImageNet-pretrained ResNet-101 to extract $14 \times 14 \times 1024$ feature maps for MAC, MAC(P) and LCGN. For CNN-LSTM, we use the 2048-dimensional feature from the last pooling layer. The setup of MDETR is the same as the original paper with ResNet-101 as backbone. We first train only the task of part detection for 30 epochs, and then train the full PTR with question answering loss.

For NS-VQA, we use Mask R-CNN [18] to generate segmentation proposals of objects and parts, respectively. The Mask R-CNN is trained on 20% of the training data annotated with ground-truth masks for 30,000 iterations. We do not include labels of categories and attributes when training segmentation. We extract the categories and attributes of objects and parts using attribute networks (ResNet-34). The extraction works in a top-down, bottom-up fashion. First, the object proposals are sent to attribute networks to propose object categories. They are then paired with contextual information (*i.e.,* the original images), to produce pose and physics information. Second, the part proposals are augmented with top-down object proposals and categories and sent to the attribute networks, to predict the part categories and colors separately. We also output the probability of a part being a line or a plane, and the orientation (parameters of the line/plane equation) of the geometric parts. Finally, in a bottom-up fashion, if parts cover a region that is not covered by object segmentations, we reconstruct objects by taking the summed area of the parts, and label the objects by taking into account the labels of parts. The program generator of NS-VQA is trained on 1,000 question-program pairs. The part segmentations and attributes are also used in MAC(P).

| Question Type | Model | Rand. | Freq. | UP-DOWN | C-LSTM | LCGN | MAC | MAC(P) | MDETR | NS-VQA | Human |
|---|---|---|---|---|---|---|---|---|---|---|---|
| Concept | query-object | 20.0 | 26.5 | 71.8 | 73.3 | 76.4 | 76.8 | 80.5 | 79.8 | **96.7** | 95.5 |
| | exist-object | 50.1 | 52.4 | 72.6 | 73.9 | 77.1 | 74.8 | 76.5 | 78.2 | **91.5** | 97.8 |
| | count-object | 9.8 | 19.8 | 50.0 | 58.9 | 68.5 | 63.3 | 64.5 | 67.9 | **80.8** | 95.4 |
| | query-part | 8.9 | 10.9 | 34.4 | 35.7 | 31.5 | 48.2 | 49.4 | 52.3 | **71.8** | 89.1 |
| | count-part | 10.0 | 20.1 | 45.5 | 46.8 | 50.4 | 49.6 | 50.9 | 51.2 | **60.6** | 82.9 |
| Relation | spatial | 11.5 | 18.2 | 40.6 | 47.2 | 51.2 | 54.9 | 59.3 | 57.6 | **67.1** | 85.7 |
| | geometric | 40.1 | 41.7 | 48.6 | 57.9 | 58.6 | 64.8 | 65.6 | **67.1** | 55.3 | 76.4 |
| | same-attribute | 36.1 | 39.5 | 47.3 | 52.9 | 54.8 | 57.6 | 58.0 | 61.4 | **62.5** | 94.2 |
| Analogy | spatial | 26.1 | 32.9 | 19.2 | 51.0 | 54.3 | 53.4 | 53.3 | 58.5 | **68.7** | 84.2 |
| | geometric | 12.1 | 26.7 | 22.2 | 26.7 | 23.3 | 23.3 | 30.0 | **34.3** | 32.5 | 73.7 |
| Arithmetic | compare-integer | 49.7 | 51.7 | 69.7 | 74.0 | 74.5 | 75.0 | **75.3** | 72.0 | 67.1 | 83.3 |
| | sum-minus | 10.0 | 19.8 | 26.8 | 31.1 | 35.0 | 34.2 | 34.8 | 36.2 | **43.4** | 75.6 |
| Physics | stability | 37.9 | 43.5 | 50.5 | 56.3 | 60.7 | 59.2 | **61.5** | 58.0 | 41.2 | 74.0 |
| | possible-change | 40.6 | 48.1 | 53.5 | 54.1 | 57.2 | **57.8** | 56.8 | 53.8 | 52.0 | 79.8 |
| | All | 28.4 | 33.5 | 45.4 | 52.9 | 55.4 | 57.5 | 58.9 | 59.8 | **62.1** | 84.8 |
| | Supervision | / | / | A | A | A | A | A,M,S | A,M,T | M,S,P | |

Table 2: Test accuracies on the PTR dataset. For the supervision types, "A" denotes the answers, "M" denotes the segmentation masks, "S" denotes the semantics of objects and parts, "T" denotes the spans of tokens associated with masks, and "P" denotes the programs of the questions. **Though humans achieve an 84.8% accuracy, SOTA visual reasoning models struggle with it**.

## 4.2 Result Analysis

We summarize the performances for each question type of baseline models in Table 2. All models are trained on the training set until convergence, tuned on the validation set, and evaluated on the test set.

**Overall Result** From the table, we can see that the neural-symbolic model, NS-VQA, outperforms neural models by a large margin. This is credited to the extra supervision of both vision (ground-truth part masks and attributes) and language (ground-truth programs). Still, it cannot achieve nearly perfect performances in basic conceptual problems due to the difficulty in perception module (*i.e.,* mask-rcnn and attribute net). MDETR stands out from all the neural models. However, two additional ingredients are required to train MDETR: the bounding boxes of all the parts, and the alignment between vision and language. We further observe that augmenting MAC with part segmentations and attributes boost the accuracy. The results suggest that currently, extra supervision is crucial for both neural models and neural-symbolic models to perform compositional reasoning. On the contrary, if the part segmentations and representations are extracted in an unsupervised manner (*e.g.,* UP-DOWN), the results are much inferior to the results of other baseline models. We also include the unsupervised segmentation results by Slot Attention [39] in the Supplementary Material. The results suggest that our benchmark can also be served as a nature testbed for unsupervised part detection.

All baseline models have unsatisfactory performances compared to human evaluation, especially in question types that require specific reasoning skills, such as analogy, physics and arithmetic.

**Question Types** We observe a variance of performances across the wide spectrum of reasoning tasks. Starting with physics problems, we find that although NS-VQA has great performances for conceptual

questions, its performance drops drastically when it comes to physical problems, worse than all neural baselines by 15% to 20%. The same phenomenon can be observed in the geometry-type problems, where NS-VQA performs worse than most of the neural baselines. This indicates that the attribute nets in current neural-symbolic models can only learn explicit attributes while struggling with implicit high-level attributes such as physics and geometry. MAC(P) is superior to MAC in "stability" problems, indicating that the physical properties of objects are linked to the composition of parts.

As for tasks that focus on part-level reasoning, models augmented with part-level representations (MAC(P), MDETR, NS-VQA) have better performances than others. Specifically, MDETR excels in geometric problems, and NS-VQA manages to achieve good performances for geometric analogy problems while other models fail. MAC(P) outperforms MAC in query-part, count-part and geometric questions, showing that part-based representations are essential for reasoning on part-level attributes and relationships.

## 4.3 Data Efficiency

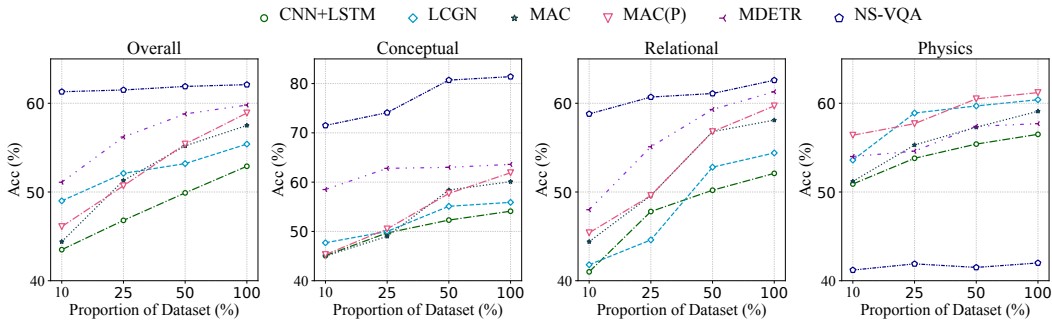

Figure 4: Compared results on data efficiency of different baseline models.

We compare different baselines on a systematic study on data efficiency, which is presented in Figure 4. Specifically, we train the model with 10%, 25%, 50% and 100% randomly chosen training images paired with questions, and evaluate the model on the original testing dataset. For NS-VQA, we use the corresponding percentage of annotated data for part detection and question parsing.

From Figure 4, we find that the neuro-symbolic model remains good performances across all splits. This might be the consequence of the disentanglement of perception module and reasoning module. LCGN is data-efficient, achieving $\sim 49\%$ accuracy with only 10% data, but does not show large improvement with more data. This might be due to the context-aware representations which provide more information with fewer data. All the models improve slightly on the physics category, which suggests that the underperformance on physics problems is rarely caused by insufficient data. Increasing the amount of data boosts the performances in the conceptual questions most, indicating that data is important for learning part-whole concepts.

## 4.4 Cross-Category Generalization

To examine machines' ability of interconnecting objects of different categories via common parts (*i.e.,* generalizing to unseen categories based on seen categories), we curate a subset from the original dataset. The curated training set has only three object categories: chair, refrigerator, and cart. The testing set contains objects of all five categories, and questions asking about the common parts of the categories (*e.g.,*, both the table and the refrigerator have doors). The cross-category training set has about 15% data of the original training set. For fair comparison, we extract approximately the same amount of data in the training set that have all five categories in the scenes, denoted as "original" in Figure 5.

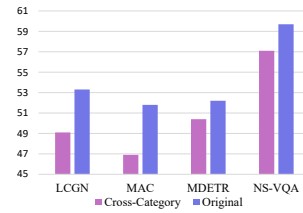

Figure 5: Results on cross-category generalization.

Figure 5 summarizes the cross-category generalization results. From the figure, we can see that NS-VQA shows great generalization ability, mainly because the part detection and part attribute extraction trained on one category can be easily transferred to another with similar part appearances. Of all the neural models, MDETR has the best performance, which is largely due to modular detection. LCGN also achieves satisfying performance. However, the cross-category accuracy for LCGN is much lower than the original accuracy, suggesting that the good performance is due to data efficiency. MAC(P) has the same problem. Although we augment image features with part segmentations, it does not show good generalization ability given by parts. One explanation is that choosing the right form of part features is crucial. Explicit mechanisms as in MDETR and NS-VQA can lead to better performances and generalization ability.

## 4.5   Ablative Studies of NS-VQA

We further eliminate perception difficulty, and investigate the challenges of our benchmark if ground-truth part-segmentations are given. We experiment with two settings: 1) Replacing masks output by Mask-RCNN with ground-truth masks (instance segmentations given); 2) Based on 1), replacing attributes output by attribute networks with ground-truth attributes (both instance and semantic segmentations given). The results are shown in Figure 6.

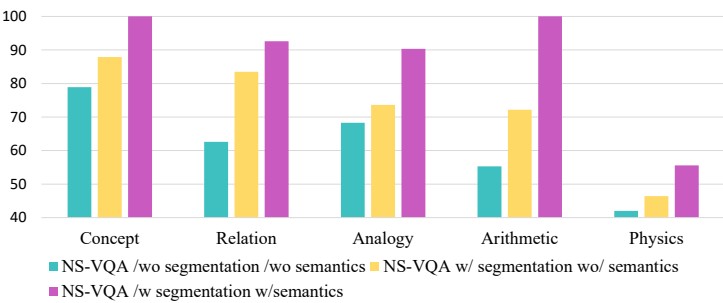

Figure 6: Ablative Studies of NS-VQA model

We can see that given all the ground-truth segmentations and semantics, NS-VQA can achieve perfect results in conceptual and arithmetic problems. However, it still performs poorly in geometric and physical problems. Given only segmentations, NS-VQA has a much lower accuracy than NS-VQA with semantics, showing that existing attribute network architectures also fall short in predicting semantics on our dataset.

## 5   Conclusion and Future Work

In this paper, we propose PTR, a novel large-scale benchmark that emphasizes visual reasoning on part-whole hierarchies. We propose several challenging question types for PTR, including: concept, relation, analogy, arithmetic and physics. We describe the detailed dataset creation procedure. We further conduct experiments on state-of-the-art visual reasoning models on PTR. Experimental results show that current visual reasoning models underperform humans in part-based reasoning tasks by a large margin. We believe this benchmark will open up opportunities for building and testing a new family of visual reasoning models.

For future work, we can leverage the depth image in PTR and experiment with RGBD-based visual perception models. Also, results show that current models fail to represent part-whole hierarchies delicately, encouraging the design of novel neural networks that specializes in this aspect (*e.g.,* Hinton [19] proposes some feasible ideas.) Last but not least, all models have bad performances in physical reasoning. One promising direction is to integrate physics engines with visual reasoning models to boost performances.

## Acknowledgments and Disclosure of Funding

This work was supported by MIT-IBM Watson AI Lab and its member company Nexplore, ONR MURI (N00014-13-1-0333), DARPA Machine Common Sense program, DSTA, ONR (N00014-18-1-2847) and Mitsubishi Electric.

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
