# A  Basic Functions

Each question in PTR is associated with a functional program built from a set of basic functions. We detail the semantics of these functions.

## A.1  Data Types

Our basic functional building blocks operate on values of the following types:

- `Object`: A single object in the scene.
- `ObjectSet`: A set of zero or more objects in the scene.
- `PartSet`: A set of parts in the scene.
- `Integer`: An integer between 0 and 9 (inclusive).
- `IntegerSet`: A set of integers.
- `Boolean`: Yes or No.
- `Value Types`:
    - Object Category: `Chair, Bed, Table, Refrigerator, Cart`
    - Part Category: `arm, leg, back, seat, central support pedestal, leg bar, wheel, arm vertical bar, arm horizontal bar, door, sleep area, top , drawer, shelf, body`
    - Color: `gray, red, blue, green, brown, purple, cyan, yellow`
    - Stability: `Stable, Unstable`
    - Possible Change: `to_left, to_right, to_front, to_behind`
- `Spatial Relationship: left, right, in front of, behind, above, below`
- `Geometric Relationship:  line-line perpendicular, line-line parallel, plane-plane perpendicular,     plane-plane parallel,     line-plane perpendicular, line-plane parallel`

## A.2  Object-Level Functions

Object-level functions focus on object-level reasoning, and are listed in Table 3.

## A.3  Part-Level Functions

Since concepts, attributes and relationships are defined on the semantic level rather than instance level, we do not use a single `Part`. Rather, we use `PartSet` to denote both a set of parts of the same semantics, as well as a set of parts of different semantics. A dictionary keeps the correspondence between objects and parts (*e.g.,*, {obj0: [part0, part1, part2], obj1: [part3, part4]...}), which facilitates hierarchical reasoning. Part-level functions are listed in Table 4. There are also arithmetic functions which focus on arithmetic problems on the number of parts, which are listed on Table 5.

| Function | Description | Input | Output |
|----------|-------------|-------|--------|
| **scene** | Returns the set of all objects in the scene | $\emptyset$ | `ObjectSet` |
| **unique** | If the input is a singleton set, then return it as an `Object` | `ObjectSet` | `Object` |
| **relate_object** | Return all objects in the scene that have the specified spatial relation to the input object | `Object, Spatial Relationship` | `ObjectSet` |
| **count_object** | Returns the size of the input set | `ObjectSet` | `Integer` |
| **exist_object** | Returns `yes` If the input set is nonempty and `no` if it is empty | `ObjectSet` | `Boolean` |
| **filter_object_category** | Filter objects by category | `ObjectSet, Object Category` | `ObjectSet` |
| **filter_stability** | Filter objects by stability | `ObjectSet, Stability` | `ObjectSet` |
| **query_object_category** | Query the category of an object | `Object` | `Object Category` |
| **query_possible_change** | Query possible changes for the objects to stay stable | `Object` | `Possible Change` |
| **query_is_stability** | Query whether an object is stable or unstable | `Object, Stability` | `Boolean` |
| **query_is_possible_change** | Query whether applying possible change to an object can make it stable | `Object, Possible Change` | `Boolean` |
| **same_object_category** | Return the set of objects that are of the same category as the inpuy | `Object` | `ObjectSet` |
| **equal_object_category** | Check whether two categories are the same | `Object Category, Object Category` | `Boolean` |
| **query_object_analogy** | The first `Object` A and the second `Object` B have certain spatial relationships. We want to find the fourth `Object` D so that the third `Object` C and the fourth `Object` D have the same | `Object, Object, Object` | `Object` |

Table 3: Object-Level Functions

# B    Human Performance

To evaluate human performances for our dataset, we hire graduate students in a university, at a rate of \$15/hr. We first provide the subjects with detailed concepts from PTR dataset, and sample questions and answers for the subjects to get familiar with the dataset. We present 2500 random questions from the test set, and take a majority vote among three subjects for each question.

# C    NS-VQA details

For NS-VQA, we first use Mask-RCNN to propose segmentations for objects and parts. Next, we extract attributes using attribute networks, which have the same parameters as in [56]. Object segmentations are sent to the attribute net to predict the categories of the objects. To facilitate reasoning on questions about spatial relationships, we follow [56], augment the objects with the original images, and use the attribute net to predict the coordinates of each object. Object proposals augmented with original images are also used to predict the stability (stable, unstable) of each object. If an object is unstable, possible changes (to_left, to_right, to_front, to_behind) are predicted.

And then, the part proposals are augmented with the predicted object proposals and attributes, to predict the part attributes. Specifically, if the part is within the region of an object proposal, we limit the categories of the parts to the specific part categories of the object categories (*e.g.,*, if the predicted object is cart, we limit the part categories to body and wheel). The part proposals are also augmented with the object proposals and fed to the attribute networks. To predict geometric information, the

| Function | Description | Input | Output |
|---|---|---|---|
| **expand_parts** | Return the parts of a set of objects. | ObjectSet | PartSet |
| **relate_part** | Return the set of parts that have the certain geometric relationship to the partset | PartSet, Geometric Relationship | PartSet |
| **exist_part** | Return Yes if the PartSet is not empty | PartSet | Boolean |
| **filter_part_category** | Filter parts by category | PartSet | PartSet |
| **filter_part_color** | Filter parts by color | PartSet | PartSet |
| **count_part** | Count the number of certain parts in each object. Specifically, using the object-part dictionary, count how many parts of each object are in the PartSet, return as IntegerSet. IntegerSet has the same size as the ObjectSet returned by **scene** | PartSet | IntegerSet |
| **filter_part_exist** | Filter the set of objects that contain parts in the PartSet | PartSet | ObjectSet |
| **filter_part_count** | Filter a set of objects that have a number of parts that matches the input Integer. Specifically, **count_part** returns the number of parts in the PARTSET of each object as IntegerSet. **filter_part_count** takes IntegerSet and filters objects by checking whether the corresponding number of parts in the IntegerSet matches the input Integer. | IntegerSet | ObjectSet |
| **query_part_category** | query the category of parts | PartSet | Part Category |
| **query_part_color** | query the color of parts | PartSet | Color |
| **same_part_color** | Return the set of parts with the same color as input | PartSet | PartSet |
| **equal_part_category** | Check whether two categories are the same | Part Category, Part Category | Boolean |
| **equal_part_color** | Check whether two colors are the same | Color, Color | Boolean |
| **query_part_analogy** | The first PartSet A and the second PartSet B have certain spatial relationships. We want to find the fourth PartSet D so that the third PartSet C and the fourth PartSet D have the same relationships. | PartSet, PartSet, PartSet | PartSet |

Table 4: Part-Level Functions

| Function | Description | Input | Output |
|---|---|---|---|
| equal_integer | Check whether two integers are equal | Integer, Integer | Boolean |
| greater_than | Check whether the first integer is greater than the second | Integer, Integer | Boolean |
| less_than | Check whether the first integer is smaller than the second | Integer, Integer | Boolean |
| sum | Return the sum of two integers | Integer, Integer | Integer |
| minus | Return the difference between two integers | Integer, Integer | Integer |

Table 5: Arithmetic Functions

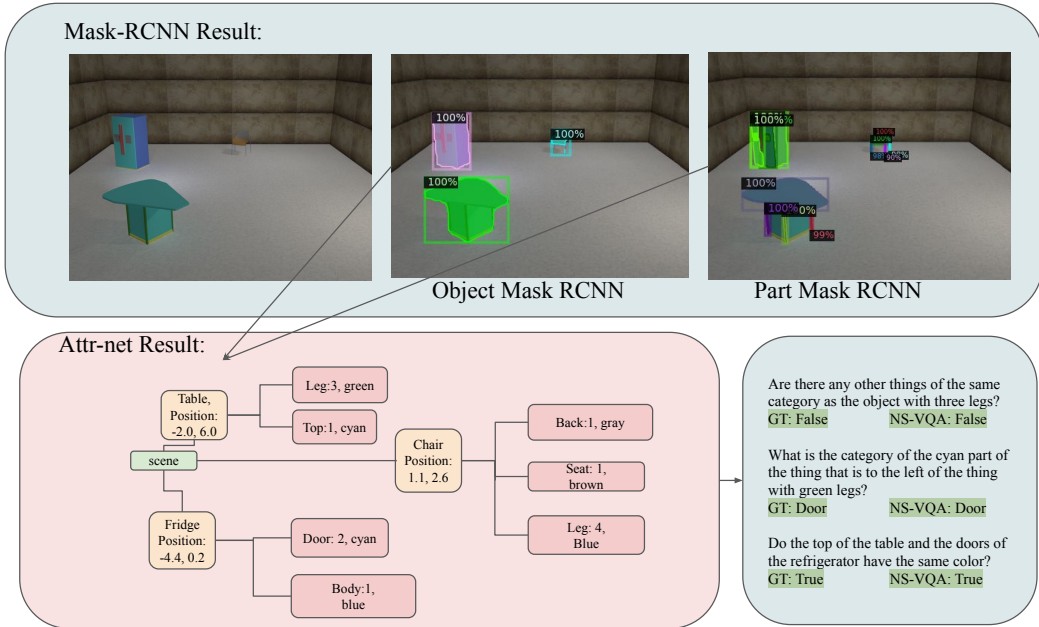

Figure 7: An example of NS-VQA.

attribute network first outputs probabilities suggesting whether the parts can be considered lines or planes. If true, the network outputs a three-dimensional vector, which denotes the parameters of the line/plane equation. If a part proposal is not within the region of an object proposal, we first check whether it's connected to an object proposal, or that the distance between the part proposal and the object proposal is smaller than a threshold. If so, we add the region of the part proposal to the object proposal, treat the part as a part within the region of an object proposal, and predict the attributes of the parts as above. If the part is not within, connected to or close to an object proposal, we first augment the part proposal with a bounding box that is greater than the part proposal and covers the part proposal, to represent the contextual information. The part proposal is then sent to the attribute network to predict its attributes. The part categories are not limited in this case. Then, we group these isolated part proposals if they are connected to or close to each other in a threshold. We take the output probabilities of part categories. We calculate the probabilities of object categories by summing up the maximum probabilities of part categories that belong to the corresponding object categories. We determine the categories of the objects using these probabilities, and then determine the part categories according the object categories and the probabilities of part categories.

In Figure 7, we provide an example of the inputs and outputs of NS-VQA.

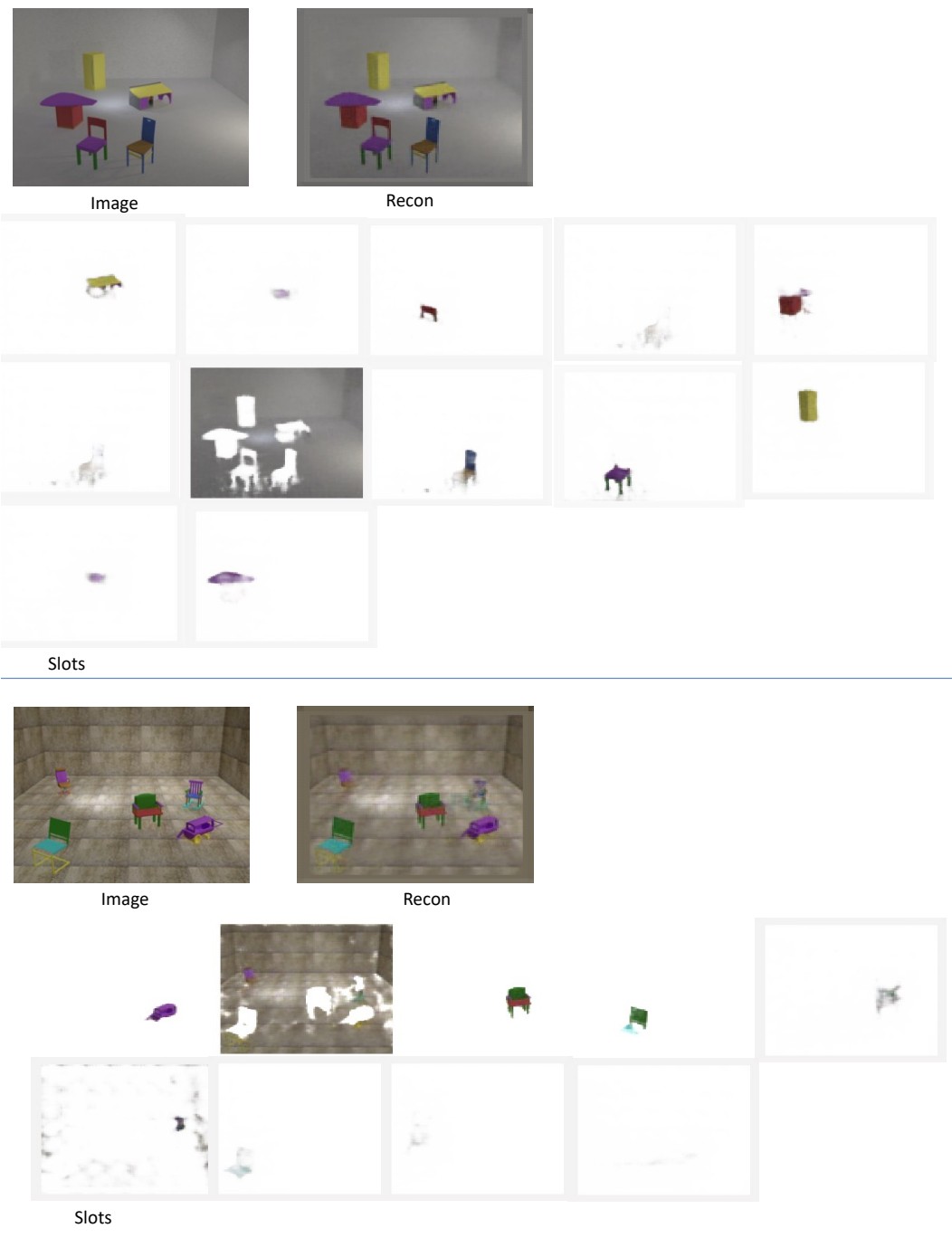

Figure 8: Slot Attention Results

# D   Unsupervised Part-Centric Rrepresentations

Here we show unsupervised segmentation results provided by Slot Attention[39], which is shown in Figure 8.

**Concept**

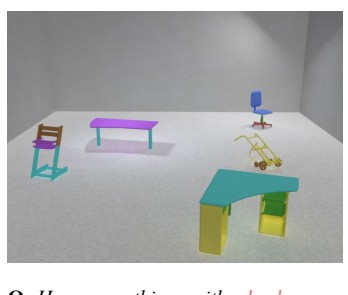

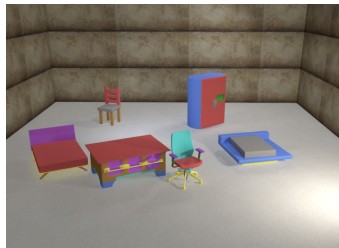

**Q:** *How many things with wheels are there?*
**A:** *2*
**Q-type:** *count_object*

**Q:** *How many legs does the object with one central support have?*
**A:** *5*
**Q-type:** *count_part*

**Q:** *How many beds with cyan back are there?*
**A:** *2*
**Q-type:** *count_object*

**Q:** *Are there any objects with two cyan legs?*
**A:** *Yes*
**Q-type:** *exist_object*

**Q:** *What is the color of the sleep area of the bed with brown legs?*
**A:** *Red*
**Q-type:** *query_part*

**Q:** *What is the category of the thing with legs?*
**A:** *Table*
**Q-type:** *query_object*

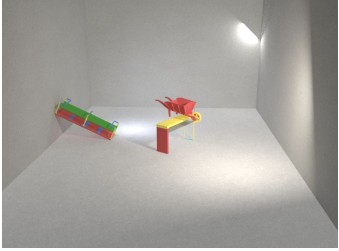

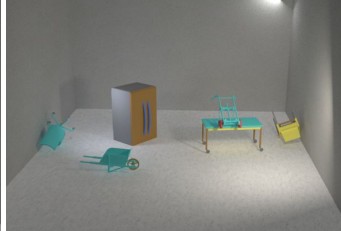

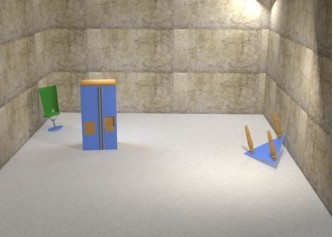

**Q:** *How many tables with shelves are there?*
**A:** *2*
**Q-type:** *count_object*

**Q:** *What is the color of the top of the table with brown legs?*
**A:** *cyan*
**Q-type:** *query_part*

**Q:** *What is the color of the pedestal of the chair?*
**A:** *Blue*
**Q-type:** *query_part*

**Q:** *Are there any objects with yellow legs?*
**A:** *Yes*
**Q-type:** *exist_object*

**Q:** *How many doors does the refrigerator have?*
**A:** *2*
**Q-type:** *count_part*

**Q:** *How many legs does the table have?*
**A:** *3*
**Q-type:** *count_part*

Figure 9: Exemplar images and questions on conceptual problems

# E  Examples

Randomly chosen exemplar images, questions and answers are shown in Figure 9-12. We categorize the questions using the semantics types of the questions. We also show the sub-type of each question, denoted as Q-type in the figures.

**Relation**

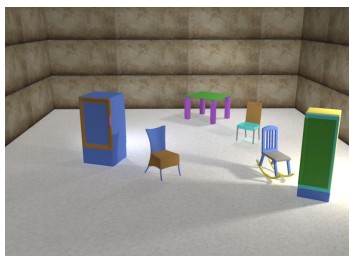 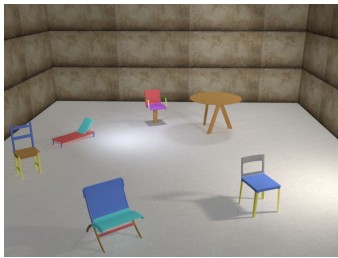 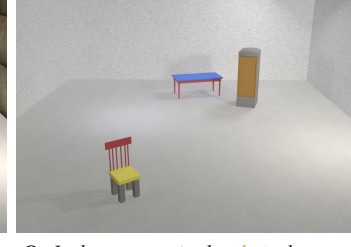

**Q:** *What is the number of the visible legs of the thing that has the same color of back as the chair with one gray seat?*
**A:** *3*
**Q-type:** *same_relate*

**Q:** *What is the category of the object with brown legs that is on the right side of the chair with a pedestal?*
**A:** *Table*
**Q-type:** *spatial_relationship*

**Q:** *Is there a part in the chair that can be considered a line, and is parallel to the brown part of the refrigerator?*
**A:** *No*
**Q-type:** *geometric_relationship*

**Q:** *Are the object with purple legs and the object with gray legs of the same category?*
**A:** *No*
**Q-type:** *same_relate*

**Q:** *What is the category of the green part of the object with four visible legs that is on the left side of the chair with two arm vertical bars?*
**A:** *Leg bar*
**Q-type:** *spatial_relationship*

**Q:** *What is the color of the part in the chair that can be considered a plane, and is perpendicular to the brown part of the refrigerator?*
**A:** *Yellow*
**Q-type:** *geometric_relationship*

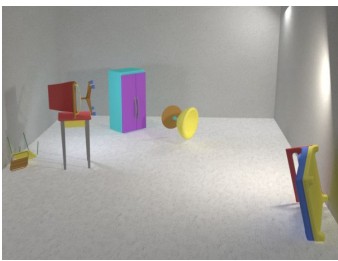 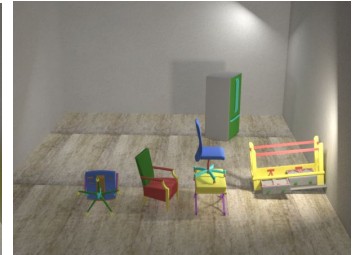

**Q:** *What is the color of the legs of the thing that has the same color of back as the bed?*
**A:** *Brown*
**Q-type:** *same_relate*

**Q:** *How many things with legs are in front of the refrigerator?*
**A:** *5*
**Q-type:** *spatial_relationship*

**Q:** *Are the body of the refrigerator, and the central support of the chair with pedestal of the same color?*
**A:** *Yes*
**Q-type:** *same_relate*

**Q:** *What is the color of the leg bars of the thing with yellow legs that is to the right of the table with yellow top?*
**A:** *Red*
**Q-type:** *spatial_relationship*

Figure 10: Exemplar images and questions on relational problems

**Analogy**

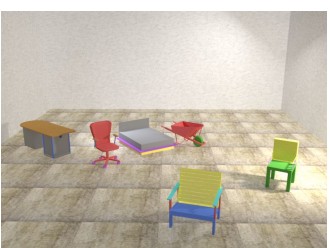 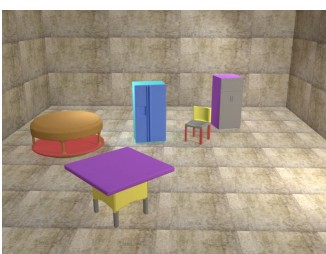 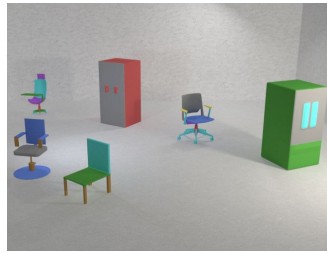

**Q:** *The thing with five legs has certain positional relation to the object with blue seat. By analogy, how many objects does the bed have the same positional relation to ?*
**A:** *2*
**Q-type:** *positional analogy*

**Q:** *The cart has certain positional relation to the chair with central support. By analogy, is there an object that the chair with green seat has the same positional relation to?*
**A:** *Yes*
**Q-type:** *positional_analogy*

**Q:** *The purple part of the table with three gray legs has certain geometric relation to the red part of the table with one brown top. by analogy, the blue part of the refrigerator with cyan body has the same geometric to the part of which color in the refrigerator with purple body?*
**A:** *gray*
**Q-type:** *geometric_analogy*

**Q:** *The yellow parts of the chair with gray back have certain geometric relation to the gray part of the refrigerator with green body. By analogy, the brown parts of the chair with green seat have the same geometric relation to a part of what color in the chair with cyan legs?*
**A:** *Blue*
**Q-type:** *geometric_analogy*

**Arithmetic**

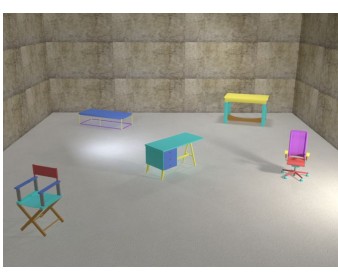

**Q:** *What is the sum of the legs in the table with blue top, and the number of drawers in the table with cyan top?*
**A:** *9*
**Q-type:** *sum-minus*

**Q:** *What is the number of arm vertical bars in the chair with brown legs, subtracted by the number of drawers in the table with cyan top?*
**A:** *1*
**Q-type:** *sum-minus*

**Q:** *Are there an equal number of doors in the refrigerator, and leg bars in the chair with cyan seat?*
**A:** *Yes*
**Q-type:** *compare-integer*

**Q:** *Are there fewer legs in the chair with gray seat, than legs in the chair with yellow seat?*
**A:** *No*
**Q-type:** *compare-integer*

**Q:** *Are there fewer legs in the table with purple top, than drawers in the table with cyan legs?*
**A:** *No*
**Q-type:** *compare-integer*

**Q:** *What is the number of drawers in the table with cyan legs, subtracted from the number of legs in the table with purple top?*
**A:** *2*
**Q-type:** *sum-minus*

Figure 11: Exemplar images and questions on analogical and arithmetic problems

**Physics**

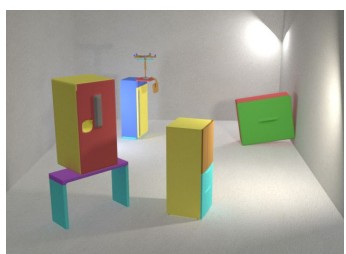 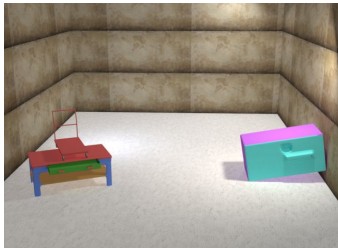 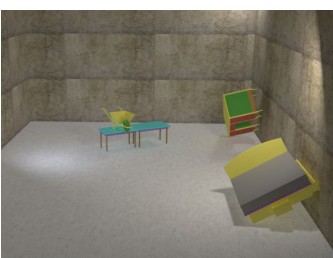

**Q:** *Is the chair stable?*
**A:** *No*
**Q-type:** *stability*

**Q:** *Towards which direction should the chair move to become stable?*
**A:** *To left*
**Q-type:** *possible changes*

**Q:** *How many objects are stable?*
**A:** 1
**Q-type:** *stability*

**Q:** *Can moving front makes the cart stable?*
**A:** *no*
**Q-type:** *possible changes*

**Q:** *Is the table with yellow top stable?*
**A:** *No*
**Q-type:** *stability*

**Q:** *Is the bed stable?*
**A:** *Yes*
**Q-type:** *stability*

Figure 12: Exemplar images and questions on physical problems