# OpenReview forum: "PTR: A Benchmark for Part-based Conceptual, Relational, and Physical Reasoning"
_NeurIPS.cc/2021/Conference — NeurIPS 2021 Poster_

### Official Review · Reviewer_XMc6 · 2021-06-29

**Rating:** 7
**Confidence:** 5

**Summary:**

This paper proposes a benchmark for part-based commonsense reasoning. The dataset consists of realistic-looking scenes with various objects in the scene. Several queries are designed to test a model's reasoning capability which includes concepts, relations, physics, analogy, and arithmetic. The author(s) run several baselines spanning from a random picking to an oracle model (NS-VQA) which utilizes ground truths for vision model and language programs.

**Main Review:**

Strengths:
1) A strong aspect of the dataset is the "analogical reasoning" queries which is an essence of commonsense reasoning. This is an underexplored area in the literature and there hardly exists any realistic dataset that captures this aspect (RAVEN only covers this aspect to simple 2D shapes as the authors have pointed out).
2) In general, the presentation of the paper is neat and easy to follow. The proposed parts-based dataset is significantly novel covering major aspects of commonsense reasoning.


Weaknesses/questions:
3) Related works:  GSGN [Ref1] seems to be closely related work. GSGN can extract the part-whole hierarchy in the form of a tree. The authors also propose a Compositional CLEVR dataset which although is less realistic as compared to PTR, but does showcase its performance on downstream tasks.

4) All the baselines either use the complete image or the ground truth object representations. A baseline with self-supervised object-centric representations like [Ref2] can be a measure of where those works lie in the spectrum of fully supervised (oracle) to distributed representations (CNN+LSTM).

5) The general trend of baseline models (including oracle NS-VQA) performing poorly in "geometric type questions" as compared to the human baseline is observed. Is this because of NOT considering 'depth' as input? (since the geometric relationships as shown in the example images (in supp. and main paper) mostly deal with figuring out the relation between lines and planes where depth plays a crucial role). I do not think an explicit experiment is required for this, but intuitive reasoning would suffice.

Suggestions:

6) For easy readability, it would be better to include the types of data each model in Table 2 takes for training (example: MDETR uses bounding boxes, LCGN is self-supervised, etc.)

References:

[Ref1] Generative Scene Graph Networks, Fei Deng et al, ICLR 2021.
[Ref2] Language-Mediated, Object-Centric Representation Learning, Ruocheng Wang and Jiayuan Mao et al, ACL 2021.


Justification for rating: I'm currently giving the paper a weak accept. If the author(s) are able to address the comments in the weaknesses section convincingly, I'm more than happy to increase my rating to an accept.


------
Edit (post-rebuttal): After carefully going over the author(s) rebuttal for *all* the reviewers, I have increased my score to 7. The author(s) have successfully addressed all the concerns I've had.

**Time Spent Reviewing:**

6 hours

---

> ### Author Response · Authors · 2021-08-10
> **Response to Reviewer XMc6**
>
> Thank you for your insightful and constructive comments.
>
> > **Q1: GSGN [Ref1] seems to be closely related work.**
>
> Thanks for pointing out this recent work. [Ref1] is definitely a very nice work and is highly related to our work. We would love to include detailed discussions of this paper in the related works.
>
> The compositional-CLEVR dataset shares the same merits with us in that it represents part-whole relationships. However, the compositional-CLEVR dataset contains only primitive shapes such as cubes, while our benchmark is more natural since it gives indoor scenes which are composed of real-life objects and their parts.
>
>
> > **Q2: A baseline with self-supervised object-centric representations like [Ref2] can be a measure of where those works lie in the spectrum of fully supervised (oracle) to distributed representations (CNN+LSTM).**
>
> Good question! We actually experimented with [Ref2] (LORL) before. We use Slot Attention as our perception module. We found the accuracy is only slightly better than random baselines due to its poor perception. We also experimented with a model where we use the unsupervised segmentation module as in [Ref2], but replace the quasi-symbolic reasoning module with bottom-up and top-down attention [Ref3] (Up-Down). The results of the models are shown below:
>
> **Table2**
>
> | Method      | All  | Concpet| Relation | Analogy | Arithmetic | Physics|
> | ----------- |:-----------:|:-----------:|-----------:|-----------:|-----------:|-----------:|
> |LORL| 34.6| 35.1| 32.0| 33.9| 37.5| 45.5|
> Up-Down| 50.3| 52.6| 51.3| 38.3| 52.4| 53.1|
>
>
> Here are the detailed analyses:
> * **[Poor perception performance]** The unsupervised segmentation by Slot Attention (SA) or MoNet used in [Ref2] is quite inaccurate on our dataset. [Ref2] uses some specified designs, such as painting the individual legs with separate colors, which is tailored for their model. However, if the legs are painted with the same color, SA or MoNet can not separate the legs as different parts. Moreover, they only use chairs, the parts of which are apart from each other and have salient geometric shapes. In other categories, some parts are even harder to be separated because they are highly embedded in or merged with other parts (e.g., the doors or the drawers of the refrigerator and the table). Moreover, the hierarchical structure of objects and parts make it harder to detect all objects and parts.
> *  **[Poor symbolic reasoning performance due to bad perception]** The performance of the quasi-symbolic reasoning module in [Ref2] highly depends on the part-centric representations given by SA. Since the perception module is pretty bad, it cannot provide any meaningful part representations. Therefore, it will give the reasoning module a cold start, and there’s little chance the perception module could improve by simply adding the QA loss.
> *  **[Additional baseline using a neural model with unsupervised part representations]** To bypass the low performance by quasi-symbolic reasoning, we then replace the symbolic QA module with a softer neural QA module, and still use the unsupervised object representations provided by SA. Here, we did an experiment using [Ref3] (Up-Down), where we use the slot features output by SA as the bottom-up attention features, and perform top-down QA reasoning. The accuracy is lower than all neural baselines in the paper. This further proves that the incorrect representations may have an adverse effect on the reasoning performances.
> *  **[Essential future work]** However, we consider it an essential future work to improve unsupervised part-centric representations, especially when part-whole hierarchies are involved. Our benchmark can also serve as a nature testbed for this purpose.
>
> > **Q3: The general trend of baseline models (including oracle NS-VQA) performing poorly in "geometric type questions" as compared to the human baseline is observed. Is this because of NOT considering 'depth' as input?**
>
> Thanks for your questions.  As stated in Line 363 in the future works, we indeed plan to experiment with RGBD-based visual perception models in the future. Here are some of our insights regarding how RGBD images might improve the performances on geometric questions.
> * **The segmentation results can be more accurate**. 2D segmentations can be inaccurate when two parts are very close in the image even though they might be distant in the 3D space. The inclusion of depth images could separate the parts more clearly, making the segmentation task easier.
> *  **The attribute network that predicts 3D geometric attributes can be more accurate**. Currently, the attribute network takes in the 2D segmentations of the parts. It’s hard to predict the 3D geometric attributes such as 6D poses from this 2D part. It’s possible for a part to tint, but still look the same in 2D image from a certain camera viewpoint.
> * **How to utilize prior knowledge as humans remains a challenge.** It might be easier for humans to perform well in geometric type questions, because humans possess certain prior knowledge about the structures and geometries of the objects. For example, it might be hard for humans to predict the geometry of a leg given the 2D part segmentation, but they can derive the part geometry by symmetry/ rotation of other parts (e.g., they know that five legs in a computer chair are spread out with equal intervals, and the total angles sum up to 360). For machines, challenges remain as to how to learn such prior knowledge, and how to incorporate the prior knowledge into neural networks.
>
> > **Q4: For easy readability, it would be better to include the types of data each model in Table 2 takes for training**
>
> Thanks for your suggestion. We will add the supervision types in the Table.
> &nbsp;
>
> &nbsp;
>
> [Ref1] Generative Scene Graph Networks, Fei Deng et al, ICLR 2021.
>
> [Ref2] Language-Mediated, Object-Centric Representation Learning, Ruocheng Wang and Jiayuan Mao et al, ACL 2021.
>
> [Ref3] Bottom-Up and Top-Down Attention for Image Captioning and Visual Question Answering, Peter Anderson, Xiaodong He, Chris Buehler, Damien Teney, Mark Johnson, Stephen Gould, Lei Zhang, CVPR 2018.
>
> *Please let us know if you have any further questions for our paper, or would like us to explore more into some new experiments.*

---

> > ### Comment · Reviewer_XMc6 · 2021-08-19
> > **Thank you for the rebuttal. Some minor follow up questions.**
> >
> > 1) I agree with the authors that GSGN only focusses on datasets with primitive 3D shapes (and also mostly with similar sizes).
> >
> > 2) Thank you for running the experiment on LORL and giving a detailed analysis of it during the rebuttal.
> >
> >     a) I'd like to further know where exactly does SA/MoNet give wrong object masks? Say in case of a chairs, is clustering of all 4 legs into same object slot the problem? Or are there several issues in segmentation as well (such as inaccurate segmentation during heavy occlusions or interference of other objects segments and/or background in the mask). I'm aware of such problem in spatial slot methods like SPAIR and SPACE so was wondering where do these models currently fail for such relatively complex datasets?
> >
> >      b) I agree that if the perception module has poor object-centric masks and hence representations, then it is very intuitive that the reasoning performance would be poor.
> >
> >      c) Iff possible, a single image and their slot segmentation/masks can be provided in an anonymous link.
> >
> >      d) Lastly, this seems to be a very good experiment and I'd like the authors to include this in the final version of the paper as well, if accepted.
> >
> > 3) Predicting geometric attributes explicitly would surely help in better reasoning performance. Also, being able to learn a 3D world model of the scene can also help imagine the scene from a different viewpoint and hence help in reasoning. Thanks for giving a detailed explanation to this.
> >
> > 4) Also the inclusion of NS-VQA(I) and NS-VQA(IS) is very interesting (in rebuttal to reviewer tBaK). The poor performance in geometric and physics based tasks do showcase the current limitations of existing works as well.
> >
> > Having read the other reviews and the rebuttal of the authors as well, I vote for an accept of the paper and I am increasing my score to *"7: Good paper, accept"*. I once again thank the authors for their extensive experimentation during the rebuttal time and clear explanations.

---

> > > ### Author Response · Authors · 2021-08-23
> > > **Response to Reviewer XMc6**
> > >
> > > Thank you. We are really inspired by your insightful comments.
> > >
> > > > **Iff possible, a single image and their slot segmentation/masks can be provided in an anonymous link.**
> > >
> > > We provide examples in [neurips7750.github.io](https://neurips7750.github.io). The first 3 examples are results on our PTR dataset. Example 1 helps explain our parameter choice. Example 2 shows a scene with interfering objects. Example 3 shows a scene with a more complex background. Example 4 and Example 5 are results on the PartNet-Chairs dataset of [Ref1], where the parts of the chairs are colored differently. Example 6 and Example 7 are results on the original PartNet dataset, where all parts have the same color.
> > >
> > > > **I'd like to further know where exactly does SA/MoNet give wrong object masks? Say in case of a chairs, is clustering of all 4 legs into same object slot the problem? Or are there several issues in segmentation as well (such as inaccurate segmentation during heavy occlusions or interference of other objects segments and/or background in the mask). I'm aware of such problem in spatial slot methods like SPAIR and SPACE so was wondering where do these models currently fail for such relatively complex datasets?**
> > >
> > > We analyze the results of the examples in detail, which is shown below.
> > >
> > > * **[Number of slots]** We would first like to clarify the choice of the number of slots for our conducted experiments. We choose 16 to be the number of slots. We use *Example 1* to explain why we choose this number. We experiment with 7 slots, 16 slots and 24 slots. We can see that with 7 slots, only object-level segmentations are provided, while our dataset focuses on part-level reasoning. With 16 slots, the objects can be divided into sub-parts, but the parts are not as fine-grained as what we want. Then we increase the number of slots to be 24. However, no finer-grained parts are detected, and the additional slots are pure-white images. Therefore, we choose 16 as the number of slots in the experiments.
> > >
> > > We then analyze the reasons for errors in detail.
> > > * **[Part-level error: merging parts]**
> > >
> > >      **PTR** From the segmentations in *Example 1* (16 slots), we can see that SA can detect some parts (*e.g.*, the table top in Slot 1 and the back in Slot3). However, it tends to merge some parts together (*e.g.*, in Slot 9 it merges the seat with four legs, and in Slot 4 it merges the back with the seat).
> > >
> > >      **PartNet-Chairs** *Example 3* and *Example 4* show the results on the PartNet-Chairs dataset of [Ref1]. We can see that the same problem exists. If the legs, wheels and the central support are painted in the same color, SA cannot distinguish them (*e.g.*, *Example 3*). However, if the four legs are painted in different colors, SA manages to separate them (*e.g.*, *Example 4*).
> > >
> > >      **PartNet** *Example 5* and *Example 6* further reveal this problem with merging legs.
> > >
> > > * **[Object-level error: objects interfering with each other make the part detection harder]**
> > >
> > >     For your question that asks about the interference of objects, we show the object segmentations of a scene with stacking objects in *Example 2* of the link. SA can roughly distinguish the cart from the table. However, the cart seems to divide the table in the middle (Slot 15). The parts in the overlapping area cannot be detected and are ambiguous.
> > >
> > > * **[Scene-level error: interference of the background]**
> > >
> > >     It's known that unsupervised segmentation methods tend to fail when the background is complex. *Example 3* shows a scene where the textures of the walls and the floor are more complex. We can see that some objects cannot be detected at all from the background, while a few others leave some of their parts in the background slot (Slot 11).
> > >
> > > The above analyses show that the unsupervised segmentations by current models are far from satisfying, especially when parts are involved. Our dataset indeed provides a challenge for unsupervised segmentations where both objects and parts are involved.
> > >
> > > > **This seems to be a very good experiment and I'd like the authors to include this in the final version of the paper as well, if accepted.**
> > >
> > > We will surely explore more into the unsupervised setting, include these experiments and add more analyses in the revision.
> > >
> > > [Ref1] Language-Mediated, Object-Centric Representation Learning, Ruocheng Wang and Jiayuan Mao et al, ACL 2021.
> > >
> > > &nbsp;
> > >
> > > *Thanks again for your insightful suggestions and comment. We would really appreciate it if you could raise your rating. Please do not hesitate to contact us if there are other clarifications or experiments we can offer.*
> > >
> > > *Thank you for your time!*
> > >
> > > Best,
> > > Authors

---

> > > > ### Comment · Reviewer_XMc6 · 2021-08-23
> > > > **final decision from my end**
> > > >
> > > > Thank you for sharing the anon link with different examples. The examples make it more clear so as to where the unsupervised Slot  attention module fails.
> > > >
> > > > Also, I've increased the rating to 7 in my original review (I'd mistakenly forgotten to do so previously).

---

### Official Review · Reviewer_jSyT · 2021-07-11

**Rating:** 7
**Confidence:** 3

**Summary:**

This paper mainly proposed a new dataset named ParT Reasoning (PTR) dataset for reasoning the part of the objects in an image, which data is synthetic from the simulator rather than the real world. The dataset is a form of visual question answering that containing five types of questions, including concept, relation, analogy, arithmetic, and physics. The authors conducted several popular methods on their PTR dataset to validate the challenge and characteristics of the dataset.

**Limitations And Societal Impact:**

The benchmark is based on the simulation, which exists a gap between the synthetic data and real-world data.

There is only one word for each answer.

**Main Review:**

Strengths:
1. It is an interesting research topic for mining 3D objects by hierarchical part-based reasoning.
2. The definition of question types is well-defined to cover six cognition-related aspects.
2. The paper is well organized.
3. Sufficient experiments to compare with other SOTA methods.

Weakness:
I wonder whether the hierarchical scene graphs to be used as shown in Figure 1(center) in the experiments. What do the effect of these scene graphs bring in the part-based conceptual, relational, and physical reasoning?

2. In Figure 4 (Physics part), I observed the NS-VQA achieved the worst performance compared with others, which doesn't match the description in Line 321.

3. In Line 348, the author claimed the MDETR has the best performance. In my opinion, the NS-VQA should be the best performer in Figure 5. Please correct me if I'm wrong.


**Time Spent Reviewing:**

10 hours

---

> ### Author Response · Authors · 2021-08-10
> **Response to Reviewer jSyT**
>
> We appreciate the positive and constructive comments from you! We will modify our paper according to your comments.
>
> > **Q1: What do the effect of these scene graphs bring in the part-based conceptual, relational, and physical reasoning?**
>
> The scene graphs are used both in data generation and the NS-VQA model.
>
> * **[Data generation]** When generating data, we have ground-truth scene graphs, so that questions can be asked based on the information provided by the scene graphs. Programs are also defined on the objects and parts of the scene graphs. If we execute the programs on the scene graphs, we can get the final ground-truth answer.
>
> * **[NS-VQA]** For NS-VQA model, instead of using ground-truth scene graphs, we use Mask-RCNN and attribute net to extract the objects and parts. We can construct new scene graphs using the methods described in Supplementary C. Also, instead of ground-truth programs, we predict programs using LSTM. The predicted programs are executed on the predicted scene graphs to generate the predicted answers.
>
> > **Q2: In Figure 4 (Physics part), I observed the NS-VQA achieved the worst performance compared with others, which doesn't match the description in Line 321.**
>
> Thank you for pointing it out. In Line321, we mean that the performances of the neural-symbolic model do not change much with regard to the amount of data, showing its data efficiency. We realize we make an unclear statement here. We will rewrite the sentence.
>
> > **Q3: In Line 348, the author claimed the MDETR has the best performance. In my opinion, the NS-VQA should be the best performer in Figure 5.**
>
> In Line348, we mean that the MDETR is the best among purely neural models, while NS-VQA is a neural-symbolic model.
>
> > **Q4: The benchmark is based on the simulation, which exists a gap between the synthetic data and real-world data.**
>
> * Our goal is to provide a synthetic diagnostic dataset similar to CLEVR [1] and CLEVRER [2], which de-emphasizes the perception realism but provides a better controlled test-bed for part-based reasoning.
>
> * As in CLEVR [1], the merit of a synthetic dataset is bias control and diagnostic ability. Real-world data often contain noise and question-conditioned bias. Thus, with real-world data, it’s hard to diagnose which parts of a model go wrong, and models tend to utilize the bias of the dataset to achieve good performances. Synthetic data allows us to control the distribution of scenes, questions and answers. Moreover, we can diagnose models with ground-truth scene graphs and programs.
>
> * More generally, our dataset represents a bet that synthetic and diagnostic dataset will enable progress in visual reasoning. It is hard to know for sure how well the resulting systems will eventually transfer to the real-world, but it is a bet that many in the community see as worth making!
>
> > **Q5: There is only one word for each answer.**
>
> If the answers are long and open-ended, a language generation model needs to be used to generate the answer. We hope that a large suite of current visual reasoning models can be evaluated on our benchmark. Therefore, we keep a limited dictionary of answers. The answer generation process can be implemented using scoring rather than language generation. The latter may bring a lot of uncertainty and noise to the diagnosis of the reasoning process.
>
> *We sincerely appreciate your comments. Please feel free to let us know if you have further questions.*

---

### Official Review · Reviewer_tbaK · 2021-07-12

**Rating:** 6
**Confidence:** 4

**Summary:**

This paper proposed a new benchmark for visual reasoning on part-whole hierarchies, where the dataset contains 80k RGBD synthetic images with ground truth object and part level annotations. It contains five types of questions including, concept, relation, analogy, arithmetic, and physic. The dataset generation procedure is described in a detailed way. Several vanilla baselines, CNN-based baselines, and neuron symbolic baseline are included and compared.

**Limitations And Societal Impact:**

The major concern of the reviewer is how much further effort do we need to answer the questions if we already know the part segmentations (instance & semantic)? For all types of questions except the physical type, it seems we can easily answer them based on part segmentations. For example, performing arithmetic is summing up semantically related parts, and performing analogy is comparing two specific parts. Will the proposed benchmark and dataset is highly dependent on the 3D segmentation backbone? If you replace the Mask RCNN with ground-truth annotations, what performance will NS-VQA achieve?

The reviewer is not sure about the quality of the design data. SCENE A: The topology of a scene is easy to tell if the shapes are separated from each other. SCENE B: But, it is hard to distinguish the topology of scenes if the shapes are superimposed with each other via stange postures. Also, such topology may be very rare and can not help predict other topologies. If the dataset contains a lot of easy SCENE A or SCENE B, it may be a little bit meaningless to perform part reasoning in a part-hierarchy way.

For the cross-domain section, the reviewer believes the generalisability lies in the shared parts for different shape categories as the author said in #346 - #347.  Therefore, a more salient way is probably training on a category and testing on a similar category and a dissimilar category? The reviewer is not sure about the similarity between refrig. and bed or cart and table, where currently your training categories contain refrig. and cart.

**Main Review:**

This paper takes a step further on part-level and probably part-whole hierarchies understanding, where the reviewer believes it is a very important direction. It would help us to study the part-hierarchies in the context of 3D parts reasoning.

The dataset is generated via knowing 3D parts and pre-defined templates. If the templates are designed well, the image is supposed to be perfect aligned with corresponding questions as well as answers. Also, the 3D shapes and parts would do not have overlaps or incomplete, and thus provides full 3D information for tested algorithms and let them focus on the part reasoning aspects.

The most intriguing thing is studying physics and found that the neural symbolic baseline failed to beat the NNs which just memorized the data. Overall the paper is easy to follow.

**Time Spent Reviewing:**

2

---

> ### Author Response · Authors · 2021-08-10
> **Response to Reviewer tbaK**
>
> Thank you very much for the insightful comments!
>
> >**Q1: How much further effort do we need to answer the questions if we already know the part segmentations**
>
> That's a very good question! We follow your instructions, and run experiments on variations of VS-VQA. One is NS-VQA(I), where ground-truth segmentations are given, but the semantics (*e.g.*, category, color) of the parts are unknown. The other one is NS-VQA(IS), where both ground-truth segmentations and semantics are given (the categories and colors are given, but positional, geometric and physical properties are unknown). Table 1 summarizes the performances compared with the original NS-VQA.
>
> **Table1.** Performance of NS-VQA without ground-truth part segmentations, NS-VQA with part segmentations but without semantics of the parts (NS-VQA(I)), and NS-VQA with both part segmentations and part semantics (NS-VQA(IS)).
>
> |      | Concept | Spatial Relationship | Geometric Relationship | Spatial Analogy | Geometric Analogy | Arithmetic | Physics |
> | ----------- |:-----------:|:-----------:|:-----------:|:-----------:|:-----------:|:-----------:|:-----------:|
> | NS-VQA | 83.1 | 77.5 | 66.2 | 69.8 | 40.4 | 61.4 | 43.4
> | NS-VQA(I)  | 87.9 | 83.9 | 69.2 | 78.9 | 42.1 | 72.2 | 55.4
> | NS-VQA(IS) | 100.0 | 98.1 | 74.6 | 97.7 | 51.3 | 100.0 | 64.9
>
> * **[Analysis of results]** We can see that given all the ground-truth segmentations and semantics, NS-VQA(IS) can achieve perfect results in conceptual and arithmetic problems. It achieves nearly perfect results in the Spatial Relationship problems, because we still need to learn the positions of the objects. However, it still performs poorly in geometric and physical problems. This shows that geometric and physical reasoning are challenging visual reasoning tasks since models with ground-truth perceptions still fail on these tasks. NS-VQA(I) has a much lower accuracy than NS-VQA(IS), showing that existing attribute network architectures also fall short in predicting semantics on our dataset.
>
> * We want to emphasize that the neural-symbolic model is simply a **rule-based oracle model**. It takes much supervision: the ground-truth segmentations and semantics, the annotated programs, as well as how to execute the programs. It remains unclear how to build a model that can learn from raw image and language, without so many human priors. We believe that PTR will open up new opportunities to many under-explored but very fundamental visual reasoning challenges.
>
> >**Q2: The reviewer is not sure about the quality of the design data. If the dataset contains a lot of easy SCENE A or SCENE B, it may be a little bit meaningless to perform part reasoning in a part-hierarchy way.**
>
> Our dataset contains both SCENE A and SCENE B. We think both scenes are meaningful in design.
>
> > **Q2.1: SCENE A: The topology of a scene is easy to tell if the shapes are separated from each other**
>  * We purposely ensure that no objects overlap in SCENE A mainly for two reasons: first, overlapping objects would result in many parts occluded, making it difficult for part segmentation; second, overlapping objects make it hard to distinguish the parts of overlapping objects.
>  * The reasoning of our tasks goes beyond reasoning about which parts belong to which object. The more difficult part is to perform reasoning on part-part geometric relationships and object-object physical relationships. Therefore, even if the objects are separated from each other, it’s not trivial to perform reasoning on scene A.
> * As stated in Line156-158, one intriguing design of our dataset is that the training and testing dataset do not share shapes. While humans have no difficulty generalizing to novel shapes, this makes scene A more difficult for machines since the novel shapes introduce unseen geometries and part-part relationships.
>
> > **Q2.2: SCENE B: But, it is hard to distinguish the topology of scenes if the shapes are superimposed with each other via strange postures.**
> * The postures are restricted and general across all splits. For the scenes with stacking objects (one shape superimposed with another), the relative positions and orientations of the two objects are restricted in a pre-defined range to avoid collisions, ensure recognition of objects, and ensure the physical reasoning is neither too hard nor too simple. For example, Scene I of Figure 1 in the paper shows a chair superimposed with a table. To design such table-chair scenes, we pre-define four possible postures: the chair is upside down and the seat touches the top of the table; 2)the chair stands on the top of the table; 3)the chair is put down 90 degrees and put on the table; 3)the chair rotates to a pose that the back and the seat, and the top of the table form a triangle. The contact area of the table and the chair varies in a range between where the chair and the table only touches a little, and where the chair has all its contact parts completely on the table. Therefore, given that we have ~80k data and only a few pre-defined postures, all postures are seen and are not rare.
>  * Even though the postures and orientations are not rare, the shapes in the test split are different from the shapes in the training split. This may make some physical scenes “unseen” or “rare”. However, this design ensures that the models cannot perform physical reasoning by simply memorizing the data or topologies. When humans see objects superimposed with each other, they can imagine the future states of the objects via the compositionality of scenes and mental simulation. It’s essential that machines can perform the same kind of mental simulation.
>
> >**Q3: For the cross-domain section, the reviewer believes the generalisability lies in the shared parts for different shape categories. A more salient way is probably training on a category and testing on a similar category and a dissimilar category?**
>
>  * **[Generalization lies in the shared parts.]** The seen categories are: chair, refrigerator and cart. The unseen categories are: table, bed. We assume that we do not know the semantics of the unseen categories, but we know some shared parts between the seen and the unseen categories. For example, chair and table both have legs and leg bars, and refrigerator and table both have doors. Test questions are about these shared parts. We want to test whether a model can transform its reasoning abilities on seen categories to unseen categories, by discovering the similarities across categories. If the test shapes are very dissimilar to the training shapes in that they do not share parts, no generalization can be reasoned about, and thus no meaningful questions can be asked.
>  * **[Multiple categories ensure the variety of questions.]** If the training scenes contain shapes of only one single category, the scenes would lack variety. Many object-object and part-part relationships cannot be defined on the scenes, and thus a lot of question templates would be invalid in this setting.
>
> *We wish that our response has addressed your concerns, and turns your assessment to the positive side. If you have any more questions, please feel free to let us know during the rebuttal window. Thank you very much! We appreciate your suggestions and comments! Thank you!*

---

> > ### Comment · Reviewer_tbaK · 2021-08-22
> > **Response**
> >
> > Thanks to the author for providing more details. The reviewer is now more confident in the dataset quality. The reviewer would like to raise the score to 6. However, submitting the paper to data track is probably better than the main conf.

---

### Official Review · Reviewer_Qvcs · 2021-07-16

**Rating:** 6
**Confidence:** 4

**Summary:**

The paper introduces a novel benchmark for part-based reasoning, featuring 80k indoor scenes contraining pieces of furniture from the PartNet dataset. Scenes are annotated with object and part labels, as well as geometric relations. Based on these features, a dataset of questions are generated as a VQA benchmark. Several existing models are evaluated on this benchmark, and are found to underperform  human subjects, sometimes substantially.

**Limitations And Societal Impact:**

Limitations and societal impact are adequately addressed.

**Main Review:**

### Dataset
The general setup of the dataset is plausible and follows previous benchmarks such as CLEVR. The main additions are the physics simulator, the geomatric features, and the new question types. These all seem like useful additions.

The most questionable choice to me is to color all parts in bright primary colors: This makes any part segmentation task very easy, but can sometimes make it difficult even for humans to determine which part belongs to which object, as there is little visual continuity between parts belonging to the same object. Exploring a version of the dataset featuring natural textures might be a good idea.

### Clarity
The text is generally clear and easy to follow. Some paragraphs, e.g. those on the conceptual importance of part based reasoning, appear longer than necessary. The discussion of the possible reasons for the performance of the evaluated methods remains rather speculative, as the methods are not modified or ablated to confirm the suspected reasons for their performance. The text mostly makes this clear, but e.g. line 322 appears very confident without clear evidence.

Minor comments:
 - Hinton's "Mapping part-whole hierarchies into connectionist networks" [15] is cited in two odd places: Line 26 uses it as psychological evidence that humans parse visual scenes into part-whole hierarchies, even though the paper contains no original psychological research. Line 31 cites it as a recent effort to represent part-whole hierarchies using neural networks, even though it was published in 1990.
 - Line 75: It is not clear why the non-synthetic nature of VQA implies great bias.
 - Line 332 is broken.
 - No "et al." should be used in line 365, as [16] is a single author paper.

### Significance
Given that the paper only introduces a new dataset, and does not propose any novel algorithm or model, determining
whether this contribution is of sufficient significance will be key to the acceptance decision. It should be noted that the
proposed dataset is composed entirely of existing freely available assets, and (aside from the human performance estimates) no
completely novel data has been collected.

Similarly, the existing methods have been applied as-is, and no modifications to their internal structure have been proposed or attempted. As a result, I am not sure if much insight regarding their inner workings can be derived from the results.

### Summary
While the proposed dataset fills a reasonable niche, it features few entirely new ideas, and the analysis of existing methods remains rather shallow. Given this, my impression is that this work does not meet the significance bar for the main track of NeurIPS. It may be better suited to the datasets & benchmarks track.

### Update
After reading the other reviews and the authors' responses, I am increasing my score from a 5 to a 6. It has become clearer to me that both the creation of the dataset and adapting the existing methods to the part-based setting involved a considerable amount of original work.

**Time Spent Reviewing:**

4

---

> ### Author Response · Authors · 2021-08-10
> **Response to Reviewer Qvcs**
>
> We thank the reviewer for the constructive comments. We will modify our paper according to these comments.
>
> >**Q1: The proposed dataset is composed entirely of existing freely available assets, no completely novel data has been collected.
> We agree that the benchmark is composed of 3D assets from PartNet. However, we respectfully push back that “no completely novel data has been collected”. We summarize our major contributions to the dataset collection below.**
> * **We propose an entirely new diagnostic dataset for part-based visual reasoning.** Over the years, we've seen that the introduction of the synthetic diagnostic datasets (*e.g.*, CLEVR[1]) has served as the major driving force for the progress of visual reasoning. However, CLEVR dataset is not challenging anymore, since a large suite of models can achieve nearly perfect performances. Therefore, it's time to step into next-level visual reasoning. Instead of reasoning on simple objects like cubes or spheres, we propose part-based reasoning on the more complex 3D objects. We believe this new perspective is not trivial and requires a deep understanding of this field.
>
> * **We propose several novel and underexplored question types.** The new question types (*e.g.*, geometry, analogy, arithmetic and physics) capture the essence of human intelligence, but have not been well investigated in previous visual reasoning datasets. The importance of these types is also acknowledged by Reviewer XMc6 and Reviewer tbaK. The results also show that such questions types are indeed challenging for state-of-the-art visual reasoning models.
>
> * **The data generation engine is fundamentally different from previous works.** Unlike CLEVR which randomly places objects in the scene, our scene generation process is much more complex. First, the positions and orientations of the objects need to be specifically calculated, so that the objects and parts can be easily recognized, there are no unrealistic scenes and the physical reasoning is neither too hard nor too simple. Second, a **physics engine** is integrated to simulate the future states of the objects, as well as possible changes to the objects to make them stable. Third, the geometric properties are extracted by sampling the point clouds of the 3D objects, deriving the equations and rejecting the parts that cannot be considered as geometric primitives. There are indeed technical contributions for building this benchmark.
> * **Questions and programs need to be carefully designed.** First, 58 templates with natural language sentences and functional programs are manually defined for the 5 questions types. The programs contain deeper and longer logic chains than previous datasets, due to the introduction of part-level reasoning and analogical reasoning. Second, bias control is harder for our benchmark since some parts are rarer than others. Strict rejection sampling is applied to control the distribution and avoid shortcuts.
>
> >**Q2: the existing methods have been applied as-is, and no modifications to their internal structure have been proposed or attempted. As a result, I am not sure if much insight regarding their inner workings can be derived from the results.**
> * **[Modifications of the models]** We did apply modifications to the models since existing methods are not really part-centric.
>
>      For the neural-symbolic model, we modify it into a bottom-up top-down hierarchical model so that it can capture the object-part relations, which is explained in detail in Supplementary Material C. We also modify the model to predict geometric and physical features. For the MAC model, we design a MAC\(P\) model which takes in the part features as additional inputs, to see whether part-level features can boost performances.
> * **[Insights from the results]** We analyze the state-of-the-art visual reasoning models of various structures on all types of questions.
>
>     Though being modified to better capture the object-part relations or part features, existing methods still have poor performances on our benchmarks. We provide detailed analysis as well as insight regarding why these methods fail, which also shows that our benchmark indeed poses a great challenge.
> * **[Additional result analyses]** Following your suggestion, we provide additional experimental results and in-depth analyses of NS-VQA model in the rebuttal.
>     1. We experiment with NS-VQA with ground-truth segmentations and semantics respectively and provide detailed analyses, which are shown in Q1 in response to Reviewer tbaK.  This further disentangles the inner structures of the NS-VQA model, and possibly provides an upper bound accuracy where ground-truth scene graphs are given.
>     2.  We experiment with settings where segmentations are trained with no supervision and provide detailed analyses, which are shown in Q2 in response to Reviewer XMc6. This demonstrates the difficulty of perception without supervision.
>     3.  For your comment:
>         > The discussion of the possible reasons for the performance of the evaluated methods remains rather speculative, as the methods are not modified or ablated to confirm the suspected reasons for their performance.
>         >
>         We will include more ablative studies and analyses like these in our modified paper. Meanwhile, we will make the language less confirmative. Thank you for your advice!
>
> >**Q3: Design choice: color all parts in bright primary colors**
>
> >**Q3.1: part segmentation task very easy**
> * **[Disentanglement of perception and reasoning]** We aim to simplify visual recognition and examine part-based reasoning skills since our goal is to construct a diagnostic dataset that focuses on part-based conceptual, relational and physical reasoning with simple visual perception. If the perception is too hard, we cannot disentangle visual perception and reasoning, and the reasoning capabilities cannot be well evaluated. This follows the common design choices in the community as seen in CLEVR [1] and CLEVRER [2]. While the segmentation of parts may be simple, part-based reasoning is still challenging. The composition of the attributes and the relationships of objects and parts enable the generation of highly compositional and complex questions.
>
> > **Q3.2: difficult even for humans to determine which part belongs to which object, as there is little visual continuity between parts belonging to the same object.**
>
> * **[Tradeoff between part segmentation and part grouping]** We admit that the coloring of parts increases the difficulty of part grouping to some extent. But we would like to point out that it is still quite easy for humans to tell objects apart through hierarchical grouping in our dataset. Our choice for coloring the parts is a tradeoff between the difficulty of part segmentation and part grouping.
> * **[Designs to make part grouping simplier]** To make sure the stacking objects can be easily separated from each other, we make several careful designs. For example, the two objects must belong to different categories, and the overlapping parts cannot be of the same colors. We also adjust the positions and the orientations of the objects, so that the entire structures of both objects can be observed. Based on the above designs, we find that it’s trivial for humans to tell the objects apart. In our human study, humans can always identify which parts belong to which objects, and obtain high accuracies as in Table 2 of our paper.
> * **[Hierarchical grouping abilities of machines]** Humans may leverage the knowledge about structures of objects to do hierarchical grouping. It’s expected that the machines can acquire the same hierarchical grouping abilities.
>
> > **Q3.3: Exploring a version of the dataset featuring natural textures might be a good idea.**
> >
> * A dataset featuring natural textures though make the hierarchical grouping a bit easier, could significantly increase the difficulty for part segmentation tasks. This is not ideal in our diagnostic dataset which focuses on part-based reasoning. For example, with natural textures, it’s hard to detect the parts which are embedded in other parts, such as drawers and doors. It’s also hard to detect tiny parts, such as the wheels of the chairs.
>
> > **Q4: Hinton's "Mapping part-whole hierarchies into connectionist networks" is cited in two odd places**
> * We are sorry for these typos and  will change the second one to be [14] (Geoffrey Hinton: How to represent part-whole hierarchies in a neural network, 2021). We will replace the first one with a psychological paper.
>
> > **Q5: It is not clear why the non-synthetic nature of VQA implies great bias.**
> * For non-synthetic datasets, it’s difficult to control the distribution of scenes, questions, and answers. Therefore, models tend to just leverage spurious correlations between images and QA pairs in the training data, instead of reasoning over the visual scenes to derive an answer.
> &nbsp;
>
> &nbsp;
>
> [1]  CLEVR: A Diagnostic Dataset for Compositional Language and Elementary Visual Reasoning. Justin Johnson, Bharath Hariharan, Laurens van der Maaten, Li Fei-Fei, C. Lawrence Zitnick, Ross Girshick, CVPR2017
>
> [2] CLEVRER: CLEVRER: CoLlision Events for Video REpresentation and Reasoning. Kexin Yi, Chuang Gan, Yunzhu Li, Pushmeet Kohli, Jiajun Wu, Antonio Torralba, Joshua B. Tenenbaum, ICLR2020
> &nbsp;
>
> &nbsp;
>
> *We wish that our response has addressed your concerns, and turns your assessment to the positive side. If you have any questions, please feel free to let us know during the rebuttal window. Thank you very much! We appreciate your suggestions and comments! Thank you!*

---

> ### Author Response · Authors · 2021-08-25
> **Looking forward to your post-rebuttal feedback**
>
> Thanks again for your insightful suggestions and comments. As the deadline for discussion is approaching, we are glad to provide any additional clarifications that you may need.
>
> In our previous response, we have carefully studied your comments and added a lot more experiments and analyses to complement your suggestions. We summarize our responses with regard to the following aspects:
>
> * **We emphasize our contribution of building an entirely new part-based visual reasoning benchmark**, which might open up new opportunities for many under-explored but fundamental commonsense reasoning challenges. PTR includes several novel and challenging question types, including physical reasoning, geometric reasoning, and analogical reasoning *etc*. The data generation engine is fundamentally new as well, and can procedurally generate much more complex physical scenes and questions than previous works.
> * **We clarify the modifications of the baseline models and the insights we can get from them**. In the rebuttal, we also include **new experiments and analyses** on the ablations of the NS-VQA model, as well as an in-depth analysis of unsupervised visual representations.  We will include these new result analyses in the revision.
> * **We provide detailed explanations of our benchmark design choices**. We explain that the coloring design is beneficial for the disentanglement of the perception module and the reasoning module, which is the key to building such kinds of diagnostic datasets.
>
>
> We agree that this work might also be qualified for the NeurIPS dataset track, which has equally stringent review as the main conference.  In the meantime, we also believe that our contributions: 1) an entirely new challenge for next-generation visual reasoning;  2) new insights of future model design choice;  are supposed to attract many audiences in this broad NeurIPS community.
> &nbsp;
>
> *We hope that the provided new experiments and additional explanations have convinced you of the merits of our work. Please do not hesitate to contact us if there are other clarifications or experiments we can offer*.
>
> *Thank you for your time again!*
>
> Best, Authors

---

> > ### Comment · Reviewer_Qvcs · 2021-08-25
> > **Update**
> >
> > Thank you for your detailed response. I am increasing my score to a 6 as a result.
> >
> > One more comment regarding the coloring design choice: Some of the harder questions in the benchmark, such as the physics and geometry ones, essentially require an understanding of the 3D scene geometry. I think it is an interesting question whether this geometry would be easier to infer (for humans and/or machines) if the scenes were depicted using more realistic colors, textures, and lighting. I can't claim that I know for sure, but I suspect this might be the case, as these features provide a lot of additional information regarding e.g. surface normals. I don't think this invalidates your approach, but I think it's a good idea to keep this in mind as a potential reason for the difficulty of these types of questions.

---

### Author Response · Authors · 2021-08-10
**General Response to All Reviewers**

We thank all the reviewers for the insightful comments and constructive suggestions to strengthen our work. In addition to the response to specific reviewers, here we would like to highlight our contributions and the new experiments that we add in the rebuttal.

**[Our Contributions]**

We are glad to find out that the reviewers generally acknowledge our contributions:
* We introduce a novel benchmark for part-based reasoning and part-whole hierarchies. The benchmark contains 80k realistic-looking indoor scenes with ground truth object and part level annotations.[Qvcs, tbaK, jSyT, XMc6]
* We propose some novel, cognition-related tasks for visual reasoning, including concept, relation, analogy, arithmetic, and physics[jSyT].  The "analogical reasoning" query is an essence of commonsense reasoning.[XMc6]. the physics simulator, the geometric features seem like useful additions [Qvcs].
* We evaluate several visual reasoning models on the benchmark to validate the challenge and characteristics of the dataset. [Qvcs, tbaK, jSyT, XMc6]

**[New Experiments]**

In this rebuttal, we have added more supporting experiments following reviewers’ suggestions.
* NS-VQA with ground-truth segmentations and semantics. [tbaK]
* Reasoning with unsupervised part representations. [XMc6]

We hope our responses below convincingly address all reviewers’ concerns. We thank all reviewers’ time and efforts again!

---

### Author Response · Authors · 2021-08-18
**Thanks for all your comments and look forward to post-rebuttal feedbacks!**

Dear AC and all reviewers:

Thanks again for all of your constructive suggestions, which have helped us improved the quality and clarity of the paper!

Since the discussion phase has started for over one week, we have not heard any post-rebuttal response yet.

Please don’t hesitate to let us know if there are any additional clarifications or experiments that we can offer, as we would love to convince you of the merits of the paper. We appreciate your suggestions. Thanks!

---

### Author Response · Authors · 2021-09-02
**Summary of our rebuttal and discussion**

We genuinely thank all reviewers and ACs for their efforts and time in reviewing our paper, as well as their constructive suggestions that contribute to the improvement of our paper. We sincerely appreciate the positive 7-7-6-6 evaluation from reviewers XMc6, jSyT, tbaK,  Qvcs.

Here is a summary of our responses:
* **[Additional Experiments]** As suggested by Reviewer tbaK, we conduct experiments on the ablations on the NS-VQA models, and validate the challenges provided by PTR in geometric, physical and analogical reasoning. As suggested by Reviewer XMc6, we add two baselines where we use slot attention to obtain unsupervised part-centric representations, and perform symbolic/neural reasoning on the representations. We also provide segmentation results and failure cases in [neurips7750.github.io](https://neurips7750.github.io). The results show that the unsupervised object and part detection is far from satisfying, and is a potential future direction provided by our benchmark.
* **[Writing]** We thank reviewer Qvcs for the helpful writing suggestions. We will make the introduction to part-based reasoning more succinct, and the language in the experiment part less confirmative.
* **[Benchmark Contributions]** We agree that this work might also be qualified for the NeurIPS dataset track, which has equally stringent review as the main conference. In the meantime, we also believe that our contributions: 1) an entirely new challenge for next-generation visual reasoning; 2) new insights of future model design choice; are supposed to attract many audiences in this broad NeurIPS community.

We owe many thanks to the reviewers for their insightful suggestions which help improve our paper a lot. The additional experiments and modifications on language will be delivered in our final version.

Best,
Authors

---

### Decision · Program_Chairs · 2021-09-27

**Decision:**

Accept (Poster)

**Comment:**


Thanks for submitting your work to NeurIPS. Proposing a new benchmark for visual reasoning on part-whole hierarchies is really an impressiv and important contribution. The dataset contains 80k RGBD synthetic images with ground truth object and part level annotations as well as informations about the concept, relation, analogy, arithmetic, and physic. Several vanilla baselines, CNN-based baselines, and neuro-symbolic baseline are included and compared. All reviewers agree that this is a very solid and strong contribution. I fully agree with this sentiment. Thanks for th enice work